# Transfer Causal Learning:
# Causal Effect Estimation with Knowledge Transfer

**Song Wei** [1]  **Ronald Moore** [2]  **Hanyu Zhang** [1]  **Yao Xie** [1]  **ishikesan Kamaleswaran** [2 1]

## Abstract

A novel problem of improving causal effect estimation accuracy with the help of knowledge transfer under the same covariate (or feature) space setting, i.e., homogeneous transfer learning (TL), is studied, referred to as the Transfer Causal Learning (TCL) problem. While most recent efforts in adapting TL techniques to estimate average causal effect (ACE) have been focused on the heterogeneous covariate space setting, those methods are inadequate for tackling the TCL problem since their algorithm designs are based on the decomposition into shared and domain-specific covariate spaces. To address this issue, we propose a generic framework called $\ell_1-\texttt{TCL}$, which incorporates $\ell_1$ regularized TL for nuisance parameter estimation and downstream plug-in ACE estimators, including outcome regression, inverse probability weighted, and doubly robust estimators. Most importantly, with the help of Lasso for high-dimensional regression, we establish non-asymptotic recovery guarantees for the generalized linear model (GLM) under the sparsity assumption for the proposed $\ell_1-\texttt{TCL}$. From an empirical perspective, $\ell_1-\texttt{TCL}$ is a generic learning framework that can incorporate not only GLM but also many recently developed non-parametric methods, which can enhance robustness to model mis-specification. We demonstrate this empirical benefit through extensive numerical simulation by incorporating both GLM and recent neural network-based approaches in $\ell_1-\texttt{TCL}$, which shows improved performance compared with existing TL approaches for ACE estimation. Furthermore, our $\ell_1-\texttt{TCL}$ framework is subsequently applied to a real study, revealing that vasopressor therapy could prevent 28-day mortality within

septic patients, which all baseline approaches fail to show.

## 1. Introduction

Causal effect estimation from observational data has attracted much attention in many fields since it is crucial for informed decision-making and effective intervention design. Several unbiased estimators for average causal effect (ACE) have been proposed, e.g., the inverse probability weighted (IPW) estimator, outcome regression (OR) estimator, and doubly robust (DR) estimator, which have shown good empirical performances and strong theoretical guarantees; see, e.g., Yao et al. (2021), for a survey of those estimators. However, in the presence of limited data in the target study, there is no guarantee both empirically and theoretically. In modern applications, advanced data acquisition techniques make it possible to collect datasets from other domains, referred to as the source domains, that are related to (but different from) that of the target study. Transfer Learning (TL), which aims to boost performance in the target domain with knowledge gained from the source domain, has shown promise in this regard (Torrey & Shavlik, 2010).

Specifically, in our motivating application, Electronic Medical Records (EMRs) from two geographically adjacent academic level 1 trauma centers are available, where, according to the fitted models, the patients not only differ in the treatment assignment mechanism but also in the way they respond to treatment. Consequently, naive integration of both datasets is impractical. Given *limited* data in the target domain, it is of great interest to find a principled TL approach to integrate *abundant* data from source domain to improve the estimation accuracy of the target domain causal effect. Indeed, TL has been considered in causal inference in a different, but more straightforward manner, due to the special treatment-and-control structure. For instance, Shalit et al. (2017); Shi et al. (2019) proposed a novel NN architecture tailored to causal effect estimation by considering shared and group-specific layers in the potential outcome models for treatment and control groups. However, adapting TL techniques from the supervised learning setting to handle data integration for causal effect estimation is non-trivial,

[1] Georgia Institute of Technology [2] Emory University. Correspondence to: Song Wei <song.wei@gatech.edu>.

*Workshop on Interpretable ML in Healthcare at International Conference on Machine Learning (ICML)*, Honolulu, Hawaii, USA. 2023. Copyright 2023 by the author(s).

as it requires counterfactual information. In causal inference, this problem is solved by the aforementioned plug-in estimators (e.g., IPW, OR, and DR estimators), which involve preliminary stage nuisance parameter estimation for the propensity score (PS) and/or OR models. Hence, a natural solution is to apply data-integrative TL to the supervised nuisance parameter estimation problem and subsequently evaluate the plug-in estimators for ACE using target domain data, where the hope is to improve ACE estimation accuracy by enhancing the quality of estimated nuisance parameters, regardless of whether the ground truth ACEs are the same across both domains.

While there has been increased interest in applying data-integrative TL techniques to causal inference in the presence of heterogeneous covariate spaces (Yang & Ding, 2020; Wu & Yang, 2022; Hatt et al., 2022; Bica & van der Schaar, 2022), these methods typically fail to handle the same covariate space setting, known as the inductive multi-task transfer learning according to Pan & Yang (2010). This limitation arises from their algorithm designs, which mostly rely on domain-specific covariate spaces. To the best of our knowledge, the first and only work studying data-integrative TL for causal effect estimation under the inductive multi-task setting, referred to as the *Transfer Causal Learning* (TCL) problem, is Künzel et al. (2018). They proposed to transfer knowledge by using neural network (NN) weights estimated from the source domain as the warm-start of the subsequent target domain NN training. Despite its improved empirical performance, the theoretically grounded approach for TCL problem is still largely missing. For other related works on applying TL in causal inference, we refer readers to an extended literature review in Appendix A and a nice survey by Yao et al. (2021).

In this work, we fill this gap by presenting a generic framework for the Transfer Causal Learning problem, called $\ell_1$-TCL framework. It entails data-integrative transfer learning of the nuisance parameter and plug-in estimation for causal effect in the target domain. The transfer learning stage comprises two steps: (i) rough estimation step using abundant source domain data, and (ii) bias correction step via $\ell_1$ regularized estimation of the difference between the target and source domain nuisance parameters using target domain data. Subsequently, the estimated nuisance parameters are plugged into the unbiased causal effect estimators, including OR, IPW, and DR estimators.

Most importantly, as shown in Bastani (2021), by leveraging techniques from Lasso for high-dimensional regression, we can establish non-asymptotic recovery guarantees for the causal effect estimators when the nuisance models (i.e. PS and OR models) are parameterized using generalized linear models (GLMs) and under the sparsity assumption on the target and source nuisance parameters' difference.

This successful application of $\ell_1$ regularized TL in causal inference could inspire a potential research direction: Recently, statistics literature has witnessed a surge of theoretically grounded TL approaches due to their empirical success, and these principled approaches could be readily adapted to the novel TCL problem; for example, TL for non-parametric regression (Cai & Pu, 2022; Lin & Li, 2023) and high-dimensional Gaussian graphical models (Li et al., 2022) might be applied to causal effect estimation and causal graph discovery (Spirtes et al., 2000; Pearl, 2009), respectively. Furthermore, given that $\ell_1$ regularization not only provides strong theoretical guarantees but also enhances empirical performance in the presence of sparsity, it is natural to incorporate recently developed non-parametric PS and OR models in $\ell_1$-TCL to improve robustness to model mis-specification. Here, we show improved performance of our $\ell_1$-TCL framework using NN-based approaches (Shalit et al., 2017; Shi et al., 2019) by comparing with existing TL approaches (Künzel et al., 2018) for ACE estimation on a benchmark pseudo-real dataset (Brooks-Gunn et al., 1992; Hill, 2011). The $\ell_1$-TCL framework is subsequently applied to a real study and reveals that vasopressor therapy could prevent mortality within septic patients, which all baseline approaches fail to show.

## 2. Problem Set-Up

We study the causal inference under Neyman–Rubin Potential Outcome framework (Rubin, 1974; Splawa-Neyman et al., 1990). In this section, we briefly review IPW, OR, and DR estimators for causal effect estimation and introduce the formal set-up of our Transfer Causal Learning problem.

**Notations.** The notations used in this work follow standard conventions. Superscript $^\mathsf{T}$ denotes vector or matrix transpose, and $\|\cdot\|_p$ denotes the vector $\ell_p$ norm. We use upper case letters to denote random variables (r.v.s) and the corresponding lower case letters to denote their realizations. For asymptotic notations: $f(n) = o(g(n))$ or $g(n) \gg f(n)$ means for all $c > 0$ there exists $k > 0$ such that $0 \le f(n) < cg(n)$ for all $n \ge k$; $f(n) = \mathcal{O}(g(n))$ means there exist positive constants $c$ and $k$, such that $0 \le f(n) \le cg(n)$ for all $n \ge k$.

### 2.1. Background on causal effect estimation

Consider the tuple $(\boldsymbol{X}, Z, Y)$ in the target study, where random vector $\boldsymbol{X} \in \mathcal{X} \subset \mathbb{R}^d$ represents covariates measured prior to receipt of treatment, r.v. $Z \in \{0, 1\}$ is treatment indicator ($Z = 1$ if treated and $0$ otherwise) and r.v. $Y$ is the *observed outcome*:

$$Y = Y_1 Z + (1 - Z)Y_0.$$

Here, $Y_0$ and $Y_1$, referred to as *potential outcomes*, are the values of the outcome that would be seen if the subject were to receive control or treatment. Throughout this work, we are interested in estimating the ACE or average treatment effect, which is formally defined as:

$$\tau = \mathbb{E}[Y_1] - \mathbb{E}[Y_0].$$

In an observational study, the treatment $Z$ is typically not statistically independent from $(Y_0, Y_1)$, since the characteristics that determine the treatment assignment may also be correlated, or "confounded", with the potential outcome. To handle this problem, a common practice is to assume there are 'no unmeasured confounders' (also known as the Ignorability Assumption):

$$(Y_0, Y_1) \perp\!\!\!\perp Z \mid \boldsymbol{X}.$$

In the following, we shall continue our study under the above assumption.

**IPW estimator.** The propensity score $e(\boldsymbol{X}) = \mathbb{P}(Z = 1|\boldsymbol{X})$ is the probability of treatment given covariates and specifies the treatment assignment mechanism. Rosenbaum & Rubin (1983) showed:

$$(Y_0, Y_1) \perp\!\!\!\perp Z \mid e(\boldsymbol{X}),$$

which leads to an unbiased estimator for ACE through the inverse probability weighting: Consider $n$ samples from the target domain:

$$\mathcal{D}_i = (\boldsymbol{x}_i, z_i, y_i), \quad i = 1, \ldots, n, \tag{1}$$

and let $\widehat{e}(\boldsymbol{x}_i)$ be the estimated propensity score for $i$-th subject, the IPW estimator for ACE is:

$$\widehat{\tau}_{\mathrm{IPW}} = \frac{1}{n} \sum_{i=1}^{n} \frac{z_i y_i}{\widehat{e}(\boldsymbol{x}_i)} - \frac{(1 - z_i) y_i}{1 - \widehat{e}(\boldsymbol{x}_i)}. \tag{2}$$

**OR estimator.** An alternative unbiased estimator uses the (potential) outcome regression model:

$$m_z(\boldsymbol{X}) = \mathbb{E}[Y_z|\boldsymbol{X}], \quad z \in \{0, 1\}.$$

Given samples (1), for $z \in \{0, 1\}$, let $n_z = \#\{i : z_i = z\}$ (# represents the cardinality of a set) and $\widehat{m}_z(\boldsymbol{x}_i)$ be the fitted potential outcome for $i$-th subject, the OR estimator for ACE is given by:

$$\widehat{\tau}_{\mathrm{OR}} = \frac{1}{n_1} \sum_{z_i=1} \widehat{m}_1(\boldsymbol{x}_i) - \frac{1}{n_0} \sum_{z_i=0} \widehat{m}_0(\boldsymbol{x}_i). \tag{3}$$

**DR estimator.** The unbiasedness of IPW and OR estimators requires correct specification of the PS and OR models,

respectively. To improve the robustness to model specification, a doubly robust (in the sense that it is unbiased when either the PS model or the OR model is correctly specified) estimator is proposed. Given samples (1), the DR estimator for ACE is defined as:

$$\widehat{\tau}_{\mathrm{DR}} = \frac{1}{n} \sum_{i=1}^{n} \frac{z_i y_i - \widehat{m}_1(\boldsymbol{x}_i)(z_i - \widehat{e}(\boldsymbol{x}_i))}{\widehat{e}(\boldsymbol{x}_i)} \\ - \frac{(1 - z_i) y_i + \widehat{m}_0(\boldsymbol{x}_i)(z_i - \widehat{e}(\boldsymbol{x}_i))}{1 - \widehat{e}(\boldsymbol{x}_i)}. \tag{4}$$

For further background knowledge on the causal inference, such as why the aforementioned estimators are unbiased, we refer readers to Appendix B.1 and some nice survey studies (Lunceford & Davidian, 2004; Bang & Robins, 2005; Yao et al., 2021).

## 2.2. Set-up for Causal Transfer Learning problem

Assume we additionally observe $n_{\mathrm{s}}$ samples of the covariates, treatment and outcome tuple $(\boldsymbol{X}_{\mathrm{s}}, Z_{\mathrm{s}}, Y_{\mathrm{s}})$ from the source domain (we will refer to (1) as samples from the target domain):

$$\mathcal{D}_{i,\mathrm{s}} = (\boldsymbol{x}_{i,\mathrm{s}}, z_{i,\mathrm{s}}, y_{i,\mathrm{s}}), \quad i = 1, \ldots, n_{\mathrm{s}}.$$

In our motivating real example, $n \ll n_{\mathrm{s}}$, rendering it difficult to get an accurate ACE estimate by solely using target domain data and necessitating the use of source domain data. However, a practical issue often arises that neither the nuisance models (i.e., PS and OR models) nor the ground truth ACEs are the same between both domains, making naively merging two datasets impractical. To be precise, consider that the PS model takes the following form:

$$\mathbb{P}(Z = 1|\boldsymbol{X}) = e(\boldsymbol{X}; \beta_{\mathrm{t}}), \quad \mathbb{P}(Z_{\mathrm{s}} = 1|\boldsymbol{X}_{\mathrm{s}}) = e_{\mathrm{s}}(\boldsymbol{X}_{\mathrm{s}}; \beta_{\mathrm{s}}), \tag{5}$$

where functions $e(\cdot)$, $e_{\mathrm{s}}(\cdot)$ have known form with unknown $d_1$-dimensional nuisance parameters, i.e., $\beta_{\mathrm{t}}, \beta_{\mathrm{s}} \in \mathbb{R}^{d_1}$. Similarly, the OR model has the following form: for $z \in \{0, 1\}$,

$$\mathbb{E}[Y_z|\boldsymbol{X}] = m_z(\boldsymbol{X}; \alpha_{z,\mathrm{t}}), \quad \mathbb{E}[Y_{z,\mathrm{s}}|\boldsymbol{X}_{\mathrm{s}}] = m_{z,\mathrm{s}}(\boldsymbol{X}_{\mathrm{s}}; \alpha_{z,\mathrm{s}}), \tag{6}$$

where functions $m_z(\cdot), m_{z,\mathrm{s}}(\cdot)$ have known form with unknown nuisance parameters $\alpha_{z,\mathrm{t}}, \alpha_{z,\mathrm{s}} \in \mathbb{R}^{d_2}$. In our TCL problem, we aim to develop a principled method to integrate data from both domains to help estimate the ACE in the target domain; to help readers understand the TCL set-up and elucidate why the TCL problem is non-trivial, we present a toy example in Appendix B.2.

# 3. Parametric Approach Based on Generalized Linear Models

We begin with a simple yet popular Generalized Linear Model (Nelder & Wedderburn, 1972) parameterization of the *nuisance models* (i.e., PS (5) and OR (6) models). A GLM for r.v. $\widetilde{Z}$ with parameter $\beta$ and predictor $\widetilde{\boldsymbol{X}}$ is:

$$\widetilde{Z} \mid \widetilde{\boldsymbol{X}} \sim \mathbb{P}(\widetilde{Z}|\widetilde{\boldsymbol{X}}) = F(\widetilde{Z}) \exp\{\widetilde{Z}\widetilde{\boldsymbol{X}}^{\mathrm{T}}\beta - G(\widetilde{\boldsymbol{X}}^{\mathrm{T}}\beta)\},$$

which satisfies

$$\mathbb{E}[\widetilde{Z}|\widetilde{\boldsymbol{X}}] = G'(\widetilde{\boldsymbol{X}}^{\mathrm{T}}\beta).$$

Here, $G'(\cdot)$, known as the (inverse) link function, is the derivative of $G(\cdot)$; common non-linear choices include sigmoid link function $G'(x) = 1/(1 + e^{-x})$ on a domain $x \in \mathbb{R}$ and exponential link function $G'(x) = 1 - e^{-x}$ on a domain $x \in [0, \infty)$. The function $F(\cdot)$ is a normalizing function ensuring a valid probability distribution. Given samples $(\widetilde{\boldsymbol{x}_i}, \widetilde{z}_i)$, $i = 1, \ldots, n$, the maximum likelihood estimation (MLE) of the GLM model parameter is given by:

$$\widehat{\beta}_{\mathrm{MLE}} = \arg\min_b \sum_{i=1}^{n} -\widetilde{z}_i\widetilde{\boldsymbol{x}_i}^{\mathrm{T}}b + G\left(\widetilde{\boldsymbol{x}_i}^{\mathrm{T}}b\right).$$

## 3.1. Data-integrative transfer learning of propensity score model parameters

As the treatment indicator is binary, the GLM parameterization with link function $G'(\cdot) = g(\cdot)$ can be expressed as follows:

$$\begin{aligned}
\mathbb{E}[Z|\boldsymbol{X}] &= \mathbb{P}(Z = 1|\boldsymbol{X}) = g(\boldsymbol{X}^{\mathrm{T}}\beta_{\mathrm{t}}), \\
\mathbb{E}[Z_{\mathrm{s}}|\boldsymbol{X}_{\mathrm{s}}] &= \mathbb{P}(Z_{\mathrm{s}} = 1|\boldsymbol{X}_{\mathrm{s}}) = g(\boldsymbol{X}_{\mathrm{s}}^{\mathrm{T}}\beta_{\mathrm{s}}).
\end{aligned} \quad (7)$$

Here, the nuisance parameters $\beta_{\mathrm{t}}, \beta_{\mathrm{s}}$ have dimensionality $d_1 = d$. Without loss of generality, we consider same link functions in both domains for simplicity; however, the success of the knowledge transfer does not rely on this "same link function condition" as long as the link functions are known.

**Guarantee for knowledge transferability.** The key assumption guaranteeing the success of the knowledge transfer is the sparsity of the nuisance parameter difference $\Delta_\beta$, defined as:

$$\Delta_\beta = \beta_{\mathrm{t}} - \beta_{\mathrm{s}}. \quad (8)$$

**Definition 1** (*s*-sparse vector). A vector $u \in \mathbb{R}^d$ is said to be *s*-sparse (with $0 \le s \le d$) if this vector has at most $s$ non-zero elements, i.e., $\|v\|_0 \le s$.

Here, we argue that the treatment assignment mechanisms should be very similar across both domains, which is characterized by the *s*-sparse difference $\Delta_\beta$, since the aforementioned two trauma centers in our study are geographically adjacent and certain clinicians are affiliated with both centers.

$\ell_1$ **regularized transfer learning of the nuisance parameters.** The first stage involves two steps: (i) leveraging abundant source domain data to estimate the source parameter $\beta_{\mathrm{s}}$, which serves as a rough estimator of $\beta_{\mathrm{t}}$ due to their sparse difference, and (ii) using $\ell_1$ regularization to learn the difference $\Delta_\beta$ from target domain data, which corrects the bias of the first-step rough estimator, i.e.,

$$\widehat{\beta}_{\mathrm{s}} = \arg\min_b \frac{1}{n_{\mathrm{s}}} \sum_{i=1}^{n_{\mathrm{s}}} -z_{i,\mathrm{s}}\boldsymbol{x}_{i,\mathrm{s}}^{\mathrm{T}}b + G\left(\boldsymbol{x}_{i,\mathrm{s}}^{\mathrm{T}}b\right),$$

$$\widehat{\beta}_{\mathrm{t}} = \arg\min_b \frac{1}{n} \sum_{i=1}^{n} -z_i\boldsymbol{x}_i^{\mathrm{T}}b + G\left(\boldsymbol{x}_i^{\mathrm{T}}b\right) + \lambda_{\mathrm{PS}}\|b - \widehat{\beta}_{\mathrm{s}}\|_1.$$

Here, $\lambda_{\mathrm{PS}} > 0$ is a tunable regularization strength hyperparameter and will be selected via cross-validation (CV) in practice. Equivalently, the bias correction step can be expressed as: $\widehat{\beta}_{\mathrm{t}} = \widehat{\Delta}_\beta + \widehat{\beta}_{\mathrm{s}}$, where $\widehat{\Delta}_\beta$ is obtained by:

$$\min_\Delta \sum_{i=1}^{n} -z_i\boldsymbol{x}_i^{\mathrm{T}}(\Delta + \widehat{\beta}_{\mathrm{s}}) + G\left(\boldsymbol{x}_i^{\mathrm{T}}(\Delta + \widehat{\beta}_{\mathrm{s}})\right) + \lambda_{\mathrm{PS}}\|\Delta\|_1. \quad (9)$$

Later in Section 4, we will show that, even when $n \ll d$ in the bias correction step, with the help of high-dimensional Lasso, $\Delta_\beta$ can be faithfully recovered with theoretical guarantees. This is quite intuitive: source domain nuisance parameters can be faithfully recovered using a large amount of source domain data, whereas the sparsity assumption guarantees valid inference of the difference using target domain data via $\ell_1$ regularization.

## 3.2. Data-integrative transfer learning of outcome regression model parameters

We parameterize the OR model via linear regression for simplicity; however, our method and theory (to be presented) can be extended to handle GLM parameterization for OR model. For $z \in \{0, 1\}$, let

$$\mathbb{E}[Y_z|\boldsymbol{X}] = \boldsymbol{X}^{\mathrm{T}}\alpha_{z,\mathrm{t}}, \quad \mathbb{E}[Y_{z,\mathrm{s}}|\boldsymbol{X}_{\mathrm{s}}] = \boldsymbol{X}_{\mathrm{s}}^{\mathrm{T}}\alpha_{z,\mathrm{s}}, \quad (10)$$

where the OR model nuisance parameters have dimensionality $d_2 = d$. Similarly, the transferability guarantee comes from the assumption that the following differences:

$$\Delta_{\alpha,z} = \alpha_{z,\mathrm{t}} - \alpha_{z,\mathrm{s}}, \quad z \in \{0, 1\}, \quad (11)$$

are *s*-sparse, i.e., $\|\Delta_{\alpha,0}\|_0, \|\Delta_{\alpha,1}\|_0 \le s$. This enables us to apply the aforementioned $\ell_1$ regularized TL techniques to estimate the OR model parameters in the target domain with the help of source domain data: For $z \in \{0, 1\}$, denote $n_{z,\mathrm{s}} = \#\{i : z_{i,\mathrm{s}} = z\}$, and let $\lambda_{\mathrm{OR}} > 0$ be the tunable regularization strength hyperparameter:

$$\widehat{\alpha}_{z,\mathrm{s}} = \arg\min_{\alpha} \frac{1}{n_{z,\mathrm{s}}} \sum_{z_{i,\mathrm{s}}=z} \left(y_{i,\mathrm{s}} - \boldsymbol{x}_{i,\mathrm{s}}^{\mathrm{T}}\alpha\right)^2,$$

$$\widehat{\alpha}_{z,\mathrm{t}} = \arg\min_{\alpha} \frac{1}{n_z} \sum_{z_i=z} \left(y_i - \boldsymbol{x}_i^{\mathrm{T}}\alpha\right)^2 + \lambda_{\mathrm{OR}}\|\alpha - \widehat{\alpha}_{z,\mathrm{s}}\|_1.$$

### 3.3. Plug-in estimation for average causal effect

In the second stage, the above fitted PS and/or OR model parameters via TL techniques are plugged into the downstream IPW (2), OR (3), or DR (4) estimators, depending on the user's confidence in the PS and/or OR model specification, to get the GLM-based $\ell_1$-TCL estimate of the ACE:

$$
\begin{aligned}
\widehat{\tau}_{\mathrm{TLIPW}} &= \frac{1}{n} \sum_{i=1}^{n} \frac{z_i y_i}{g(\boldsymbol{x}_i^{\mathrm{T}}\widehat{\beta}_{\mathrm{t}})} - \frac{(1-z_i)y_i}{1 - g(\boldsymbol{x}_i^{\mathrm{T}}\widehat{\beta}_{\mathrm{t}})}, \\
\widehat{\tau}_{\mathrm{TLOR}} &= \frac{1}{n_1} \sum_{z_i=1} \boldsymbol{x}_i^{\mathrm{T}}\widehat{\alpha}_{1,\mathrm{t}} - \frac{1}{n_0} \sum_{z_i=0} \boldsymbol{x}_i^{\mathrm{T}}\widehat{\alpha}_{0,\mathrm{t}}, \\
\widehat{\tau}_{\mathrm{TLDR}} &= \frac{1}{n} \sum_{i=1}^{n} \frac{z_i y_i - \boldsymbol{x}_i^{\mathrm{T}}\widehat{\alpha}_{1,\mathrm{t}}(z_i - g(\boldsymbol{x}_i^{\mathrm{T}}\widehat{\beta}_{\mathrm{t}}))}{g(\boldsymbol{x}_i^{\mathrm{T}}\widehat{\beta}_{\mathrm{t}})} \\
&\quad - \frac{(1-z_i)y_i + \boldsymbol{x}_i^{\mathrm{T}}\widehat{\alpha}_{0,\mathrm{t}}(z_i - g(\boldsymbol{x}_i^{\mathrm{T}}\widehat{\beta}_{\mathrm{t}}))}{1 - g(\boldsymbol{x}_i^{\mathrm{T}}\widehat{\beta}_{\mathrm{t}})}.
\end{aligned}
\tag{12}
$$

## 4. Theoretical Analysis

Typically, to make valid inferences by solely using target domain data, we need a sufficiently large amount of target domain data such that $n \gg d$. However, in our setting, such an assumption does not hold; to make things even worse, we may encounter $n < d$ case. Fortunately, with the help of techniques from Lasso for high-dimensional regression, recovery guarantees can still be established when we have abundant source domain data, which only require target domain sample size $n$ to be on the order of $\log d$. In this section, we present the main results and their interpretations; complete details including the technical assumptions and proofs can be found in Appendices C, D, and E.

**Main theoretical results.** When the PS model is correctly specified and the difference is $s$-sparse, i.e., $\|\Delta_\beta\|_0 \leq s$, in the large sample limit $n, n_{\mathrm{s}} \to \infty$, consider the following *regime*:

$$n \gg s^2 \log d, \quad n_{\mathrm{s}} \gg nd^2, \tag{13}$$

By taking

$$\lambda_{\mathrm{PS}} = \mathcal{O}\left(\sqrt{\log d}\left(\frac{1}{\sqrt{n}} + \frac{d}{\sqrt{n_{\mathrm{s}}}}\right)\right),$$

we can show that, with probability at least $1 - 1/n$, the absolute estimation error is upper bounded as:

$$|\widehat{\tau}_{\mathrm{TL}} - \tau| = \mathcal{O}\bigg(\underbrace{s\sqrt{\frac{\log d}{n}}}_{\textbf{bias correction error}} + \underbrace{sd\sqrt{\frac{\log d}{n_{\mathrm{s}}}}}_{\textbf{rough estimation error}}\bigg),$$

where $\widehat{\tau}_{\mathrm{TL}}$ can be either the TLIPW estimator $\widehat{\tau}_{\mathrm{TLIPW}}$ or the TLDR estimator $\widehat{\tau}_{\mathrm{TLDR}}$ in eq. (12).

**Interpretations.** Similar to the two-stage estimation, i.e., nuisance parameter recovery and plug-in estimation for ACE, the proofs are done by plugging the non-asymptotic upper bound on the vector $\ell_1$-norm of the nuisance parameter to the absolute error bound of the downstream plug-in estimators, resulting in **the above error bound decomposition**. In particular, the bias correction term is $\mathcal{O}(s\sqrt{\log d/n})$ (which aligns with that of the classic Lasso estimator, cf. Theorem 7.1 (Bickel et al., 2009)) and dominates the rough estimation error term due to $n_{\mathrm{s}} \gg nd^2$ (13); however, according to the above error upper bound, the condition on source domain sample size can be relaxed to $n_{\mathrm{s}} \gg s^2 d^2 \log d$ to achieve consistency. Without the help of the source domain, the overall error rate will be similar to that of the rough estimation, which requires $n \gg d^2$ target domain samples to achieve a satisfying error bound (cf. Theorem 1 (Bastani, 2021)). In contrast, the abundant source domain data, characterized by $n_{\mathrm{s}} \gg nd^2$ in the considered regime (13), relaxes the requirement on target domain sample size to $n \gg s^2 \log d$ to achieve the same satisfying error upper bound.

In our proof, we invoke the Compatibility Condition (Bastani, 2021) for the sample covariance matrix, which is standard in high-dimensional Lasso literature; alternatively, as suggested in Remark 1 (Bastani, 2021), if we consider the classic Restricted Eigenvalue Condition (Bickel et al., 2009; Meinshausen & Yu, 2009; van de Geer & Bühlmann, 2009), we can prove $\ell_2$ error bound that scales as $\sqrt{s}$ instead of $s$; see Remark 1 in Appendix C on why we consider $\ell_1$ error bound for nuisance parameter estimation over the $\ell_2$ bound. Lastly, our non-asymptotic analysis shows that the error upper bound with probability at least $1 - \varepsilon$ (for any $\varepsilon \in (0,1)$) will have a $\mathcal{O}(s\sqrt{\log(1/\epsilon)/n})$ term. When we consider the probability converging to one at a polynomial rate, i.e., $\varepsilon = 1/n^\kappa$ for positive integer $\kappa$, this term will be $\mathcal{O}(s\sqrt{\log n^\kappa/n})$ and dominated by the $\mathcal{O}(s\sqrt{\log d/n})$ term in the above bound under our considered regime (13). The above result corresponds to the $\kappa = 1$ case.

**Additional results for correctly specified OR model.** When the OR model specification is correct with $s$-sparse differences $\|\Delta_{\alpha,z}\|_0 \leq s$ ($z \in \{0,1\}$), if the samples in the treatment and control groups are "balanced" in the sense that there exists a constant $r \in (0,1)$ such that, for $z \in \{0,1\}$,

$$\liminf_{n \to \infty} \frac{n_z}{n} \geq r, \quad \liminf_{n_s \to \infty} \frac{n_{z,s}}{n_s} \geq r, \qquad (14)$$

where (recall that) $n_z = \#\{i : z_i = z\}$ and $n_{z,s} = \#\{i : z_{i,s} = z\}$, then, by taking

$$\lambda_{\text{OR}} = \mathcal{O}\left( \sqrt{\log d} \left( \frac{1}{\sqrt{rn}} + \frac{d}{\sqrt{rn_s}} \right) \right),$$

for $\widehat{\tau}_{\text{TL}} = \widehat{\tau}_{\text{TLOR}}$ or $\widehat{\tau}_{\text{TLDR}}$, we can show that with probability at least $1 - 1/n$, the absolute estimation error can be upper bounded as follows:

$$|\widehat{\tau}_{\text{TL}} - \tau| = \mathcal{O}\left( s\sqrt{\log d} \left( \frac{1}{\sqrt{rn}} + \frac{d}{\sqrt{rn_s}} \right) \right).$$

Due to space consideration, complete details are deferred to the Appendix (Appx.), including the assumptions, lemmas, formal statements of the non-asymptotic theoretical guarantees, and all proofs. To help readers find the results, we provide a summary of the locations of our theories in the Appendix; see Table 1. Furthermore, the superior empirical performance of the above GLM parametric approach is verified via numerical simulation in Appendix F.

*Table 1.* Locations of all non-asymptotic results.

|        | Nuisance parameter estimation | Plug-in ACE estimation |
| ------ | ----------------------------- | ---------------------- |
| TLIPW  | Lemma 1 (Appx. C)             | Theorem 1 (Appx. C)    |
| TLOR   | Lemma 3 (Appx. D)             | Theorem 2 (Appx. D)    |
| TLDR   | Lemma 1, Lemma 3              | Theorem 3 (Appx. E)    |

# 5. A Generic Framework for Transfer Causal Learning

Inspired by the superior performance of the GLM-based parametric approach, we now extend our method into a generic framework for the TCL problem by considering arbitrary parameterization of the *nuisance model* (i.e., PS (5) and/or OR (6) models), which is called $\ell_1-\text{TCL}$ framework. This extension can benefit from improved robustness to model mis-specification, and it is motivated by a well-known observation (Tibshirani, 1996; Fan & Li, 2001; Zou & Hastie, 2005) that, in the presence of the sparsity, $\ell_1$ regularization does not only help establish theoretical guarantee but also improves the estimation accuracy when only limited data is available. Most importantly, $\ell_1-\text{TCL}$ can be applied to conditional average causal effect estimation in the presence of heterogeneous causal effect. We will begin with formally presenting the $\ell_1-\text{TCL}$ framework.

$\ell_1$**−TCL framework.** Consider arbitrary parameterization of the nuisance model with finite-dimensional nuisance

parameter $\theta \in \Theta$. Given dataset $\mathcal{D}$, suppose the estimator for nuisance parameter can be obtained as: $\widehat{\theta} = \arg\min_{\theta \in \Theta} \mathcal{L}(\theta; \mathcal{D})$, where $\mathcal{L}$ is the loss function. In our set-up, the ground truth nuisance parameters are different across both domains, i.e., $\theta_t \neq \theta_s$, and we assume their difference $\theta_t - \theta_s$ is sparse such that this difference can be estimated from the target domain using $\ell_1$ regularization to correct the bias of the rough estimator obtained from the source domain. Formally, the *nuisance parameter estimation stage* of our proposed $\ell_1-\text{TCL}$ is given by:

**Rough estimation**:
$$\widehat{\theta}_s = \arg\min_{\theta \in \Theta} \mathcal{L}(\theta; \mathcal{D}_s),$$

**Bias correction**:
$$\widehat{\theta}_t = \arg\min_{\theta \in \Theta} \mathcal{L}(\theta; \mathcal{D}_t) + \lambda \|\theta - \widehat{\theta}_s\|_1,$$

where $\mathcal{D}_s = \{\mathcal{D}_{i,s}, i = 1, \ldots, n_s\}$ and $\mathcal{D}_t = \{\mathcal{D}_i, i = 1, \ldots, n\}$ are the collections of source and target domain samples respectively, and $\lambda > 0$ is a tunable hyperparameter. In the subsequent *plug-in estimation stage*, the IPW estimator (2), OR estimator (3), and/or DR estimator (4) are evaluated using the estimated nuisance parameters above to get the $\ell_1-\text{TCL}$ estimate of the ACE.

**Non-parametric approach based on neural networks.** While there exist many recent efforts on improving robustness in causal inference, such as meta-learning (Westreich et al., 2010) (notably, super learning (Pirracchio et al., 2015)), using NN to parameterize the nuisance models (Keller et al., 2015) is the most straightforward approach due to NN's superior model expressiveness. In the following, we will consider two recently developed NN architectures: Treatment-Agnostic Representation Network (TARNet) (Shalit et al., 2017) and Dragonnet (Shi et al., 2019); we defer further details, such as their loss functions, to Appendix B.3. The implementation of the nuisance parameter estimation stage in our NN-based $\ell_1-\text{TCL}$ is straightforward: the rough estimation step follows standard NN training using source domain data; in the bias correction step, similar to eq. (9) for GLM, we will estimate the sparse difference between the target and source NN weights with zero initialization. Complete details of our NN-based $\ell_1-\text{TCL}$ can be found in Appendix G.2.

**Application.** Heterogeneous causal effect has recently drawn increasing attention in causal inference, and there have been many popular machine learning approaches, such as meta-learning (Curth & van der Schaar, 2021) and heterogeneous transfer learning (i.e., TL under the heterogeneous covariate space setting) (Bica & van der Schaar, 2022), applied to this problem. Typically, this problem is approached via the conditional average treatment (or causal)

*Table 2.* Mean and standard deviation of absolute errors of estimated ACEs over 50 trials using IHDP dataset. The primary goal is to compare three learning frameworks: we can observe that TL can help improve ACE estimation accuracy for all ACE estimators (highlighted in green for each column) and our proposed $\ell_1$-TCL yields the best in-sample and out-of-sample results (highlighted in bold font).

| In-sample | Dragonnet | | | TARNet | | |
|---|---|---|---|---|---|---|
| | IPW | OR | DR | IPW | OR | DR |
| TO-CL | $12.479_{(26.993)}$ | $0.868_{(1.47)}$ | $0.654_{(0.702)}$ | $6.85_{(6.192)}$ | $0.567_{(0.446)}$ | $0.468_{(0.364)}$ |
| WS-TCL | $6.414_{(9.667)}$ | $0.534_{(0.552)}$ | $0.572_{(0.636)}$ | $3.502_{(4.101)}$ | $0.413_{(0.313)}$ | $0.359_{(0.22)}$ |
| $\ell_1$-TCL | $6.412_{(9.664)}$ | $0.543_{(0.557)}$ | $0.58_{(0.634)}$ | $3.326_{(3.626)}$ | $0.36_{(0.312)}$ | $\mathbf{0.293_{(0.222)}}$ |

| Out-of-sample | Dragonnet | | | TARNet | | |
|---|---|---|---|---|---|---|
| | IPW | OR | DR | IPW | OR | DR |
| TO-CL | $35.352_{(75.125)}$ | $0.826_{(1.248)}$ | $2.009_{(3.397)}$ | $5.664_{(6.884)}$ | $0.671_{(0.56)}$ | $0.367_{(0.318)}$ |
| WS-TCL | $17.684_{(20.993)}$ | $0.512_{(0.586)}$ | $1.324_{(1.492)}$ | $4.204_{(6.092)}$ | $0.476_{(0.399)}$ | $0.339_{(0.289)}$ |
| $\ell_1$-TCL | $17.682_{(20.996)}$ | $0.519_{(0.615)}$ | $1.337_{(1.494)}$ | $4.039_{(4.762)}$ | $0.418_{(0.353)}$ | $\mathbf{0.308_{(0.251)}}$ |

effect (CATE) instead of ACE, i.e.,

$$\tau_S = \mathbb{E}[Y_1 | \boldsymbol{X} \in S] - \mathbb{E}[Y_0 | \boldsymbol{X} \in S],$$

which studies the causal effect within a sub-cohort of patients whose covariates lie in a target subset of the covariate space, i.e., $S \subset \mathcal{X}$.

Built on the proposed $\ell_1$-TCL, we propose a Partition-then-Transfer approach, which we call ParT, for CATE estimation. Unlike the heterogeneous transfer learning approach by Bica & van der Schaar (2022) which may require an additional dataset with a different covariate space, ParT handles the single dataset (or multiple datasets with the same covariate space) setting. Consider samples from a single dataset as in eq. (1), let $S_t \subset \mathcal{X}$ be the target subset, and the goal is to estimate $\tau_{S_t}$. ParT first partitions the covariate space into $\mathcal{X} = S_s \cup S_t$, resulting in a source-target domain partition: $\mathcal{D}_s = \{\mathcal{D}_i : \boldsymbol{X}_i \in S_s\}$ and $\mathcal{D}_t = \{\mathcal{D}_i : \boldsymbol{X}_i \in S_t\}$. Then, $\ell_1$-TCL can be readily applied to leverage knowledge gained from $\mathcal{D}_s$ to help estimate the CATE (or target domain ACE) $\tau_{S_t}$.

In practice, the target subset $S_t$ is typically defined through a binary (or categorical) covariate, resulting in a natural covariate space partition based on the corresponding labels. As the partitioned domains come from the same dataset, it is reasonable to assume the underlying treatment assignment mechanisms are similar across both domains and therefore our $\ell_1$-TCL is applicable. Nevertheless, it is important to develop a principled approach to determine whether the knowledge is transferable from the partitioned source domain. In particular, when covariate space is partitioned via a categorical covariate with three or more labels, the problem is cast as a multiple-source TL problem since there are multiple source domains; in this case, it is important to determine which source domain to include in the transfer learning. Indeed, Tian & Feng (2022) studied this multiple-source TL

problem using the $\ell_1$ regularized approach considered in this work, which we believe can help establish theoretical guarantee for ParT; however, this is out of the scope of the current study, and we leave this for future discussion. Next, we use a pseudo-real data experiment to show the effectiveness of ParT.

## 6. Pseudo-Real Data Experiment

In this experiment, we aim to show the effectiveness of ParT for CATE estimation, which also demonstrates the good performance of its building block, i.e., our $\ell_1$-TCL framework, by comparing with baseline frameworks. As ground truth causal effects are inaccessible in most real studies, we consider a commonly used pseudo-real dataset, i.e., the Infant Health and Development Program (IHDP) dataset (Brooks-Gunn et al., 1992; Hill, 2011). It includes 747 subjects (139 treated and 608 control), with 6 continuous and 19 categorical covariates (of which 18 of them are binary). We randomly pick one binary covariate (denoted by $X_{par}$) and assign subjects with labels 0 and 1 to source and target domains, respectively, resulting in $n_s = 546, n = 201$. The goal is to study the ACE in the target domain, or the CATE for the subjects with $X_{par} = 1$. Due to space consideration, additional details for the dataset, configurations, training, and results are deferred to Appendix G.

**Baseline approaches.** We compare $\ell_1$-TCL framework with two baseline learning frameworks: solely using target domain data to estimate ACE, which we call "target only causal learning" (TO-CL), and the "warm-start" TCL baseline (WS-TCL) by Künzel et al. (2018), which used the estimated NN weights in the source domain as the warm-start of the subsequent target domain NN training. For each framework, the nuisance model for PS and OR is either Dragonnet or TARNet with hyperparameters selected based

on minimum average NN regression loss on a randomly selected validation target domain dataset; the estimated nuisance parameters are subsequently plugged into IPW, OR, and DR estimators to get the estimated ACEs.

**Results.** We report both in-sample (i.e., training and validation target datasets) and out-of-sample (i.e., testing target dataset) absolute estimation errors over 50 trials in Table 2, from which we can observe that: (i) transfer learning helps improve estimation accuracy for all *ACE estimators* (we will call a specific nuisance model coupled with a specific plug-in estimator as an ACE estimator); (ii) in most cases, our proposed $\ell_1$-TCL outperforms the existing WS-TCL approach; (iii) most importantly, the best results (highlighted in bold fonts) are given by our proposed $\ell_1$-TCL framework.

Another interesting finding is that plug-in estimators based on the OR model typically perform better than PS model-based IPW estimator, potentially due to severe model misspecification of the NN-based PS model. This is consistent with the observation noted by Shi et al. (2019), who only considered OR estimator in their experiments, and may explain why NN classification cross entropy (CE) loss and mean squared error (MSE) do not serve as good hyperparameter selection criteria in our task; those results are presented in Table 6 for completeness. To further validate the effectiveness of our $\ell_1$-TCL (as well as our ParT), we report results for source-target domain partition based on another binary covariate (which yields $n_s = 642$ and $n = 105$) in Tables 7 and 8.

# 7. Real-Data Example

In this real experiment, we aim to investigate whether vasopressor therapy can *prevent* mortality within sepsis patients. Baseline approaches that only use the target domain data or naively merge both domains' data all indicate statistically significant *promoting* effect from treatment (verified by the $90\%$ confidence intervals (CI) of the ACE estimates), which clearly violates common sense. Fortunately, by leveraging our $\ell_1$-TCL framework, we can reach a reasonable conclusion that vasopressor therapy does *prevent* mortality within sepsis patients. Due to space limitation, complete details, such as patient demographics and training details, are deferred to Appendix H.

**Data description.** We construct a retrospective cohort of patients using in-hospital data from two adjacent academic, level 1 trauma centers located in the South Eastern United States in 2018. The data was collected and analyzed in accordance with an institutional review board and relevant ethics approval information will be provided if the paper is accepted. A total of 34 patient covariates comprised

of vital signs and laboratory (Lab) results are examined in this study. Patients are considered to be treated if they received vasopressor therapy, which is defined as receiving norepinephrine, epinephrine, dobutamine, dopamine, phenylephrine, or vasopressin, at any time within the 12-hour window before sepsis onset. The outcome variable is the 28-day mortality, which is a common metric used by clinicians performing observational studies on sepsis patients (Stevenson et al., 2014).

**Baseline approaches.** We choose the PS model parameterized by GLM (7) with sigmoid link function and IPW estimator for ACE estimation. We begin with TO-CL framework, i.e., without knowledge transfer, for both domains, yielding ACEs 0.12 in the target domain and 0.057 in the source domain. Even without the ground truth, those results are counterintuitive as treatment should prevent mortality (Avni et al., 2015; Wei et al., 2022). Indeed, the estimate in the source domain is almost zero, which is closer to our "believed ground truth" than that of the target domain, potentially due to its larger sample size. Naively merging two domains' data, which we call Merge-CL framework, is a tempting choice, given that two studied trauma centers sometimes share clinicians; it leads to a point estimate of 0.082, which aligns with the intuition that Merge-CL "drags" the TO-CL estimate of the target domain towards that of the source domain, as the source domain has more samples.

*Table 3.* Comparison of estimated ACEs in the real-data example: the only reasonable result is given by our proposed $\ell_1$-TCL, which indicates *inhibiting* causal effect from the vasopressor therapy to 28-day mortality in spesis patients.

| Data used | Target domain only | Both domains | |
| --- | --- | --- | --- |
| Framework | TO-CL | Merge-CL | $\ell_1$-TCL |
| Point estimate | 0.120 | 0.082 | −0.011 |
| Bootstrap mean | 0.072 | 0.130 | −0.853 |
| Bootstrap median | 0.072 | 0.120 | −0.067 |
| Bootstrap 90% CI | [0.015, 0.134] | [0.016, 0.275] | [−7.257, 1.951] |

**$\ell_1$-TCL and uncertainty quantification.** Now, we consider TLIPW estimator (12) in our $\ell_1$-TCL, and it yields a point estimate of $-0.011$, which is much closer to the "believed ground truth"; most importantly, we now reach a more reasonable conclusion that vasopressor therapy has an inhibiting causal effect on mortality in sepsis patients. Additionally, we perform bootstrap uncertainty quantification (UQ) with 200 bootstrap trials, each with 700 random samples (with replacement) from the target domain. The baseline frameworks (i.e., TO-CL and Merge-CL) all show statistically significant promoting causal effects, verified by the $90\%$ bootstrap CI, which again violates common sense. In contrast, despite the $90\%$ CI contains zero, the mean and median of bootstrap $\ell_1$-TCL causal effect estimates all sug-

gest that vasopressor therapy can prevent 28-day mortality within sepsis patients.

**Discussion.** Reliable decision-making is essential in healthcare, which is a major application of our $\ell_1$-TCL. One common approach is UQ; however, as reflected by the wider bootstrap CI for our $\ell_1$-TCL (compared to that of the baseline approaches), the performance of our $\ell_1$-TCL is sensitive to the choice of hyperparameters — oftentimes there exist bootstrap samples where the pre-selected grid does not cover the empirical optimal choice, leading to unreasonably large or small ACE estimates. It poses a practical challenge that it requires large computational resources to perform grid search for hyperparameter selection in each bootstrap trial, rendering vanilla bootstrap impractical. Currently, the most reliable estimate for drawing causal conclusions in $\ell_1$-TCL framework would be the bootstrap median, which still indicates inhibiting causal effect from the treatment.

Indeed, this highlights an important future direction, i.e., the development of a principled approach for UQ in TCL problem. For example, Juditsky et al. (2023) recently introduced a new CI construction approach for GLM using a relatively novel concentration result of vector fields. This may facilitate the construction of CI of the nuisance parameters and hence the causal effect through the unbiased plug-in estimators. This topic is outside the scope of this work, and we leave it for future study.

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

# Appendix of Transfer Causal Learning: Causal Effect Estimation with Knowledge Transfer

## Table of Contents

# A. Extended Literature Survey

## A.1. Background on transfer learning

Transfer learning (Torrey & Shavlik, 2010) has received increasing attention due to its empirical success in various fields, ranging from machine learning problems, such as natural language processing (DAUME III, 2007), recommendation systems (Pan & Yang, 2013) and computer vision (Tzeng et al., 2017), to science problems, such as predictions of protein localization (Mei et al., 2011), biological imaging diagnosis (Shin et al., 2016), integrative analysis of "multi-omics" (e.g., genomics) data (Sun & Hu, 2016; Hu et al., 2019; Wang et al., 2019), cancer image classification (Hosny et al., 2018; Sevakula et al., 2018), drug sensitivity prediction (Turki et al., 2017) and discovery (Bastani, 2021), and so on. Based on whether or not the target and source domains as well as the target and source tasks are the same, transfer learning problems can be divided several different types (Pan & Yang, 2010). Our study focuses on "Inductive Multi-Task Transfer Learning" and our $\ell_1$ regularization-based approach can be categorized as "Transferring Knowledge of Parameters". Our work only leverages a particular transfer learning technique and we refer readers to Pan & Yang (2010); Weiss et al. (2016); Zhuang et al. (2020) to comprehensive surveys on transfer learning.

## A.2. Developments of $\ell_1$ regularized transfer learning approaches

The idea of using $\ell_1$ regularization to develop theoretically grounded TL approach could date back to Evgeniou & Pontil (2004), who considered support vector machine with parameter decomposed as summation of a shared term and a task-specific term and proposed a learning algorithm by imposing $\ell_1$ regularization on the task-specific terms in all domains. Recently, this idea was applied to GLM by Bastani (2021), and this seminal work motivates several follow-up studies: Tian & Feng (2022) extended this work to multi-source TL problems, Li et al. (2022) proved minimax optimality under liner regression setting, and later on showed minimax rate of convergence for high-dimensional GLM estimation (Li et al., 2023), and so on. Our work follows this line of study and adapts the $\ell_1$ regularized TL approach proposed by Bastani (2021) to develop a theoretically grounded method for TCL, but the theoretical results may be strengthened using those aforementioned recent developments. Most importantly, it is important to recognize that our work points out a new direction on leveraging recently developed principled methods to contribute to the TCL problem.

## A.3. Connections between transfer learning and causal inference

While the causal transfer learning problem (i.e., leveraging causal inference to help with TL problems, such as domain adaption, by exploring the invariant causal relationships between both domains) has been studied in the past few years from both empirical (Zhang et al., 2015; Magliacane et al., 2018; Yang et al., 2021) and theoretical (Rojas-Carulla et al., 2018; Chen & Bühlmann, 2021) perspectives, the reverse study on adapting TL techniques to causal inference (i.e., our proposed TCL problem) starts to attract more attention recently. In particular, a line of research (Yang & Ding, 2020; Wu & Yang, 2022; Hatt et al., 2022) focuses on the handling the unmeasured confounding variables in the target observational datasets with the help of unconfounded randomized experimental source domain data, where, in its nature, only the TL approaches for heterogeneous covariate space settings are applicable. However, such experimental data is not always available in reality, and the fundamental problem of estimating causal effects under the classic no unmeasured confounding assumption receives little attention; existing works along this direction include the aforementioned "warm-start" knowledge transfer approach under our TCL setting (Künzel et al., 2018) and a special neural network architecture designed based on the shared covariate space and the domain-specific covariate spaces (Bica & van der Schaar, 2022). Here, we not only provide a theoretically grounded approach for TCL problem, but also use numerical evidence to show our proposed $\ell_1-\texttt{TCL}$ outperforms the existing warm-start method.

# B. Additional Details for Problem Set-Up

## B.1. Additional background knowledge on causal effect estimation

The gold-standard approach to estimating the causal effect is randomized controlled trials (RCT), where subjects are randomized to receive treatment or placebo (i.e., the control group). However, RCT is unethical in most studies, such as medical study. Therefore, the main question is how to estimate causal effect from observational data.

Let us recall the notations we use for the potential outcome framework (Rubin, 1974): random vector $\boldsymbol{X} \in \mathbb{R}^d$ represents covariates measured prior to receipt of treatment, $Z \in \{0, 1\}$ is treatment indicator, $Y$ is the observed outcome: $Y =$

$Y_1 Z + (1-Z)Y_0$, as well as potential outcomes $Y_0$ and $Y_1$. The ACE, which is the estimand, is defined as: $\tau = \mathbb{E}[Y_1] - \mathbb{E}[Y_0]$.

Apparently, observing $Y_0$ and $Y_1$ simultaneously is impossible, making it a tempting choice to estimate $\mathbb{E}[Y_0]$ and $\mathbb{E}[Y_1]$ using the sample average outcome in the control and treatment group and take their difference. Unfortunately, the latter estimate $\mathbb{E}[Y|Z=0] = \mathbb{E}[Y_0|Z=0]$ and $\mathbb{E}[Y|Z=1] = \mathbb{E}[Y_1|Z=1]$, which may be different from $\mathbb{E}[Y_0]$ and $\mathbb{E}[Y_1]$ since the treatment $Z$ is typically not statistically independent from $(Y_0, Y_1)$ — the characteristics that lead a subject to receive treatment may also be correlated, or "confounded" with the potential outcome.

In observational study, although $(Y_0, Y_1) \perp\!\!\!\perp Z$ is unlikely to hold, it may be possible to identify subject characteristics (or rather, some pre-treatment covariates) related to (or can affect) both potential outcome and treatment, referred to as "confounders". If we assume the covariate vector $\boldsymbol{X}$ contains all such confounders, we would have $(Y_0, Y_1) \perp\!\!\!\perp Z \mid \boldsymbol{X}$, which is referred to as "no unmeasured confounders" or ignorability assumption (Robins et al., 2000). Under this assumption, we shall have

$$\begin{aligned}
\mathbb{E}[Y|Z=1] &= \mathbb{E}\{\mathbb{E}[Y|Z=1, \boldsymbol{X}]\} = \mathbb{E}\{\mathbb{E}[Y_1|Z=1, \boldsymbol{X}]\} \\
&= \mathbb{E}\{\mathbb{E}[Y_1|\boldsymbol{X}]\} = \mathbb{E}[Y_1].
\end{aligned} \tag{15}$$

Similarly,

$$\mathbb{E}[Y|Z=0] = \mathbb{E}\{\mathbb{E}[Y|Z=0, \boldsymbol{X}]\} = \mathbb{E}[Y_0].$$

The above observations actually motivate the unbiased estimator using the outcome regression model, i.e., the OR estimator (3). Under the no unmeasured confounding assumption, the ACE $\tau$ is identifiable from observational data.

The propensity score $e(\boldsymbol{X}) = \mathbb{P}(Z=1|\boldsymbol{X})$ is the probability of treatment given covariates, which specifies the treatment assignment mechanism. Rosenbaum & Rubin (1983) showed that $(Y_0, Y_1) \perp\!\!\!\perp Z \mid e(\boldsymbol{X})$, which implies that $\mathbb{E}[I(Z=1)|Y_1, \boldsymbol{X}] = e(\boldsymbol{X})$. Therefore, we will have

$$\begin{aligned}
\mathbb{E}\left[\frac{ZY}{e(\boldsymbol{X})}\right] &= \mathbb{E}\left\{\mathbb{E}\left[\frac{I(Z=1)Y_1}{e(\boldsymbol{X})} \,\middle|\, Y_1, \boldsymbol{X}\right]\right\} \\
&= \mathbb{E}\left\{\frac{Y_1}{e(\boldsymbol{X})}\mathbb{E}[I(Z=1)|Y_1, \boldsymbol{X}]\right\} = \mathbb{E}[Y_1].
\end{aligned} \tag{16}$$

Similarly,

$$\mathbb{E}\left[\frac{(1-Z)Y}{1-e(\boldsymbol{X})}\right] = \mathbb{E}[Y_0].$$

The above observations actually motivate the application of IPW (Horvitz & Thompson, 1952) for ACE estimation and show that IPW estimator (2) is unbiased under correct PS model specification.

One common drawback of both IPW and OR estimators is that they require correct specification of the PS and OR models respectively, which is challenging in practice. To fix this issue, an augmented IPW estimator (also known as DR estimator) is proposed (Robins et al., 1994; Rotnitzky et al., 1998; Scharfstein et al., 1999) — The main idea is, by incorporating an augmented term (which is related to the OR model) in IPW, the estimator will be doubly robust. To elucidate the doubly robustness, we re-write the DR estimator (4) as follows:

$$\begin{aligned}
\widehat{\tau}_{\mathrm{DR}} &= \frac{1}{n}\sum_{i=1}^{n}\left[\frac{z_i y_i}{\widehat{e}(\boldsymbol{x}_i)} - \frac{z_i - \widehat{e}(\boldsymbol{x}_i)}{\widehat{e}(\boldsymbol{x}_i)}\widehat{m}_1(\boldsymbol{x}_i)\right] - \frac{1}{n}\sum_{i=1}^{n}\left[\frac{(1-z_i)y_i}{1-\widehat{e}(\boldsymbol{x}_i)} + \frac{z_i - \widehat{e}(\boldsymbol{x}_i)}{1-\widehat{e}(\boldsymbol{x}_i)}\widehat{m}_0(\boldsymbol{x}_i)\right] \\
&= \frac{1}{n}\sum_{i=1}^{n}\left[\widehat{m}_1(\boldsymbol{x}_i) + \frac{z_i\{y_i - \widehat{m}_1(\boldsymbol{x}_i)\}}{\widehat{e}(\boldsymbol{x}_i)}\right] - \frac{1}{n}\sum_{i=1}^{n}\left[\widehat{m}_0(\boldsymbol{x}_i) + \frac{(1-z_i)\{y_i - \widehat{m}_0(\boldsymbol{x}_i)\}}{1-\widehat{e}(\boldsymbol{x}_i)}\right].
\end{aligned}$$

Notice that:

$$\begin{aligned}
\mathbb{E}[Y_1] &= \mathbb{E}\left[\frac{ZY}{e(\boldsymbol{X})} - \frac{Z - e(\boldsymbol{X})}{e(\boldsymbol{X})}m_1(\boldsymbol{X})\right] = \mathbb{E}\left[m_1(\boldsymbol{X}) + \frac{Z\{Y - m_1(\boldsymbol{X})\}}{e(\boldsymbol{X})}\right], \\
\mathbb{E}[Y_0] &= \mathbb{E}\left[\frac{(1-Z)Y}{1-e(\boldsymbol{X})} + \frac{Z - e(\boldsymbol{X})}{1-e(\boldsymbol{X})}m_0(\boldsymbol{X})\right] = \mathbb{E}\left[m_0(\boldsymbol{X}) + \frac{(1-Z)\{Y - m_0(\boldsymbol{X})\}}{1-e(\boldsymbol{X})}\right].
\end{aligned}$$

Therefore, the DR estimator is unbiased when either the PS model or the OR model is correctly specified. Additionally, those estimators have nice theoretical properties; see, e.g., Wooldridge et al. (2002); Wooldridge (2007) for theory of IPW

estimator and Robins et al. (1994); Bang & Robins (2005) for theory of DR estimator. There are also other approaches to estimate causal effects using propensity score, such as matching; see Lunceford & Davidian (2004) for a nice survey on the use of propensity scores in causal inference and Yao et al. (2021) for a recent comprehensive survey on causal inference.

## B.2. A motivating toy example for TCL problem

To elucidate why TCL problem is non-trivial, let us consider:

$$\underline{\text{Treatment assignment}} : \quad \mathbb{P}(Z = 1 | X_1, X_2) = g(\beta_1 X_1 + \beta_2 X_2),$$
$$\underline{\text{Causal relationship}} : \quad Y = \tau Z + \alpha X_2 + \epsilon,$$

where $g(x) = 1/(1 + e^x)$ is the sigmoid function. The goal is to infer the causal effect from treatment $Z$ to outcome $Y$, given potential confounding variables $X_1$ and $X_2$; the additive noise $\epsilon$ is independent from the aforementioned r.v.s. The treatment assignment mechanism and the causal relationship are visualized in Figure 1; further experimental details such as the configurations can be found in Appendix F.

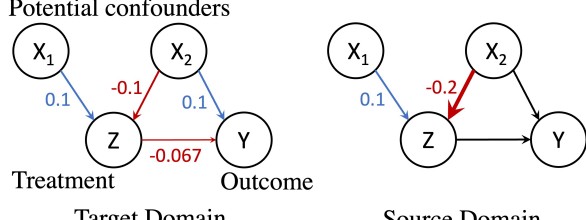

Potential confounders

Treatment        Outcome

Target Domain          Source Domain

*Figure 1.* In the toy example, the treatment assignments differ between target and source domains in that the effects from covariate $X_2$ are different. We do not impose assumptions on whether or not the ACEs are the same for both domains.

Although IPW is consistent (Wooldridge et al., 2002; Wooldridge, 2007), making inference from limited amount of target domain data leads to estimate of the ACE with large bias, as verified in Table 4. This necessitates the use of source domain data. One naive way is to integrate both datasets in the estimation of the PS model nuisance parameters. However, due to different treatment assignments, this naive data-integration will not help correct the bias. To make things even worse, since we have $n_s \gg n$, this naive data-integrative estimate will bias towards the source domain, leading to a potentially worse downstream IPW estimator, as verified in Table 4.

*Table 4.* Comparison of ACE estimation accuracy: the truth is $\tau = -0.067$. Our proposed method with knowledge transfer yields the most accurate one, which correctly recovers the *inhibiting* effect.

| Data used | Target only | Both domains | |
| --- | --- | --- | --- |
| Learning framework | TO-CL | Merge-CL | $\ell_1$-TCL |
| IPW estimate | 0.0002 | 0.0441 | $-0.0013$ |

To leverage the abundant source domain data in a principled manner, we introduce a $\ell_1$ regularized TL approach for ACE estimation, i.e., our proposed $\ell_1$-TCL framework; please see a graphical illustration in Figure 2.

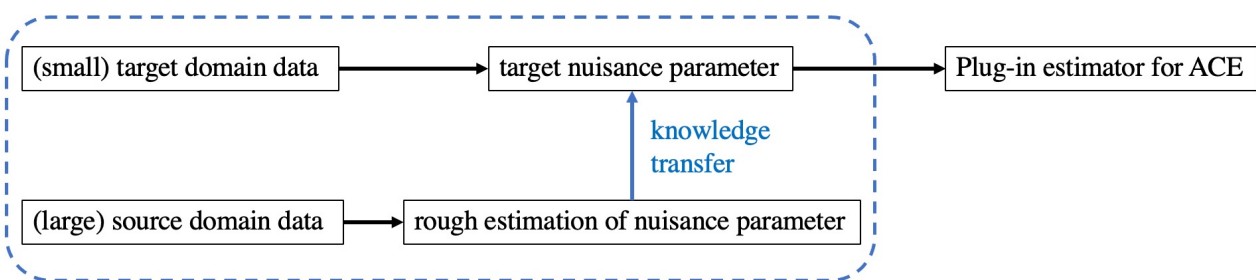

Data-Integrative Transfer Learning for nuisance parameter estimation

*Figure 2.* Illustration of the general approach for TCL problem. In our proposed $\ell_1$-TCL framework, the nuisance parameter estimation stage leverages $\ell_1$ regularized TL, and the plug-in estimation stage considers IPW, OR and DR estimators.

## B.3. Neural network-based nuisance models for causal effect estimation

In this part, we briefly review the aforementioned NN-based approaches for ACE estimation: TARNet (Shalit et al., 2017) and Dragonnet (Shi et al., 2019), which can both be categorized as representation learning method according to Yao et al. (2021).

**TARNet.** Consider covariate vector, treatment and observed outcome tuple $(\boldsymbol{X}, Z, Y)$ tuple with realizations $\mathcal{D} = \{(\boldsymbol{x}_i, z_i, y_i),\ i = 1, \ldots, n\}$. TARNet finds a representation of the covariates, denoted by $\Phi(\boldsymbol{x}_i)$ which maps the covariate vector onto a representation space, and hypothesis of the potential outcome variable, denoted by $m_{z_i}(\Phi(\boldsymbol{x}_i))$, simultaneously by minimizing the following regularized objective function:

$$\min_{m_0, m_1, \Phi}\ \mathcal{L}_{\mathrm{TAR}}(m_0, m_1, \Phi; \mathcal{D}) = \frac{1}{n} \sum_{i=1}^{n} w_i \widetilde{L}\left(m_{z_i}(\Phi(\boldsymbol{x}_i)), y_i\right)$$
$$+ \lambda_{\mathrm{CPLX}} \Re(m_0, m_1) + \lambda_{\mathrm{BAL}}\, \mathrm{IPM}\left(\{\Phi(\boldsymbol{x}_i)\}_{i:z_i=0}, \{\Phi(\boldsymbol{x}_i)\}_{i:z_i=1}\right),$$

where $\Re$ controls the model complexity, $\mathrm{IPM}(\cdot, \cdot)$ represents the Integral Probability Metric (IPM) (Sriperumbudur et al., 2012), such as the Maximum Mean Discrepancy and the Wasserstein Distance, evaluated on two empirical distributions defined by two collections of data-points on the representation space, and weights $w_i$'s compensate for the difference in treatment group size and are defined as follows:

$$w_i = \frac{z_i}{2u} + \frac{1 - z_i}{2(1 - u)}, \quad i = 1, \ldots, n, \quad u = \frac{1}{n} \sum_{i=1}^{n} z_i.$$

The loss function $\widetilde{L}$ for the network training is decomposed into two terms, i.e., $\widetilde{L}\left(m_z(\Phi(\cdot)), \cdot\right),\ z \in \{0, 1\}$, which correspond to the control and treatment groups, respectively. The weights for the treatment and control functions are updated only if the sample belongs to that group. Either the MSE or log-loss can be used as $\widetilde{L}$, depending on whether the outcome variable is continuous or binary. Most importantly, to handle the problem of variance arising from treatment imbalance, TARNet objective includes the empirical IPM to upper bound this variance; hyperparameter $\lambda_{\mathrm{BAL}} > 0$ controls the trade-off between outcome regression model fitting and the treatment-and-control distribution balanceness. When $\lambda_{\mathrm{BAL}} = 0$, it corresponds to the TARNet; otherwise, it corresponds to the Counterfactual Regression.

**Dragonnet.** Similarly, Dragonnet creates a shared representation of the covariates can be used to predict the treatment and potential outcomes. It uses a NN for the shared representation followed by two NNs used for predicting potential outcomes of the treatment and control groups respectively. However, instead of using a IPM layer, they incorporate a mapping layer for the propensity score, which is named "propensity score head" and denoted by $e(\cdot)$, to connect the shared representation of the covariates with the estimated propensity scores. To be precise, the objective function is:

$$\min_{\theta}\ \mathcal{L}_{\mathrm{Dragon}}(\theta; \mathcal{D}) = \frac{1}{n} \sum_{i=1}^{n} \underbrace{\left(m_{z_i}(\theta; \boldsymbol{x}_i) - y_i\right)^2}_{\text{NN regression loss}} + \lambda_{\mathrm{BAL}} \underbrace{\mathrm{CE}\left(e(\theta; \boldsymbol{x}_i), z_i\right)}_{\text{NN classification CE loss}}, \tag{17}$$

where $\mathrm{CE}(\cdot, \cdot)$ is the binary classification cross entropy loss and $\lambda_{\mathrm{BAL}} > 0$ is a tunable hyperparameter controlling trade-off between outcome regression model fitting and the treatment-and-control distribution balanceness.

For further details of TARNet and Dragonnet, we refer readers to the original papers. In our numerical experiments, we use the open source implementation[1] of TARNet and Dragonnet on the IHDP dataset and readers can find further implementation details therein.

## C. Non-Asymptotic Recovery Guarantee for TLIPW estimator

We begin our theoretical analysis with the TLIPW estimator. We will first prove the non-asymptotic upper bound on the $\ell_1$ regularized TL estimator for PS model and then plug it into the error bound for unbiased IPW estimator (2) to get the final recovery guarantee for the TLIPW estimator.

---

[1]The are two implementations on GitHub, one is from the Dragonnet paper author: https://github.com/claudiashi57/dragonnet, and the other is a reproduction of the results using PyTorch: https://github.com/alecmn/dragonnet-reproduced.

## C.1. Guarantee for PS model nuisance parameter estimation with knowledge transfer

Let us begin with necessary assumptions:

**Assumption 1.** The covariates in both target and source domains are uniformly bounded, i.e., there exists $M_X > 0$ such that $\|\boldsymbol{x}_i\|_\infty \le M_X, i = 1, \ldots, n$, and $\|\boldsymbol{x}_{i,\mathrm{s}}\|_\infty \le M_X, i = 1, \ldots, n_\mathrm{s}$.

The above assumption is a slightly different from the "standardized design matrix" assumption in Bastani (2021), which requires the squared matrix $F$-norms of design matrices $(\boldsymbol{x}_1, \ldots, \boldsymbol{x}_n)^\mathrm{T}$ and $(\boldsymbol{x}_{1,\mathrm{s}}, \ldots, \boldsymbol{x}_{n_\mathrm{s},\mathrm{s}})^\mathrm{T}$ to be $n$ and $n_\mathrm{s}$, respectively. However, we will see they serve the same purpose when proving Lemma 1 (to be presented). Denote the sample covariance matrices as follows:

$$\Sigma = \frac{1}{n} \sum_{i=1}^n \boldsymbol{x}_i \boldsymbol{x}_i^\mathrm{T} \in \mathbb{R}^{n \times n}, \quad \Sigma_\mathrm{s} = \frac{1}{n_\mathrm{s}} \sum_{i=1}^{n_\mathrm{s}} \boldsymbol{x}_{i,\mathrm{s}} \boldsymbol{x}_{i,\mathrm{s}}^\mathrm{T} \in \mathbb{R}^{n_\mathrm{s} \times n_\mathrm{s}}. \tag{18}$$

**Assumption 2.** The source domain sample covariance matrix $\Sigma_\mathrm{s}$ is positive-definite (PD); in particular, we assume that $\Sigma_\mathrm{s}$ has minimum eigenvalue $\psi > 0$.

Here, Assumption 2 ensures we can faithfully recover $\beta_\mathrm{s}$ using MLE from the source domain data, and this assumption is mild when $n_\mathrm{s} > d$, which is satisfied under our considered regime (13).

**Definition 2** (Compatibility Condition (Bastani, 2021)). The compatibility condition with constant $\phi > 0$ is met for the index set $\mathcal{I} \subset \{1, \ldots, d\}$ and the matrix $\Sigma \in \mathbb{R}^{d \times d}$, if for all $u \in \mathbb{R}^d$ satisfying $\|u_{\mathcal{I}^\mathrm{c}}\|_1 \le 3\|u_{\mathcal{I}}\|_1$, the following condition holds:

$$\|u_{\mathcal{I}}\|_1^2 \le \frac{\#\mathcal{I}}{\phi^2} u^\mathrm{T} \Sigma u,$$

where (recall that) $\#$ represents the cardinality of a set, and $u_{\mathcal{I}}$ is a vector with $j$-th elements being $u_j$, i.e., $j$-th element in vector $u$, if $j$ belongs to index set $\mathcal{I}$ and zero otherwise.

A standard assumption in high-dimensional Lasso literature is:

**Assumption 3.** The index set $\mathcal{I} = \operatorname{supp}(\Delta_\beta)$ (8) and target domain sample covariance matrix $\Sigma$ (18) meet the above compatibility condition with constant $\phi > 0$.

This assumption guarantees the identifiablility of $\Delta_\beta$, and it holds automatically when target domain sample covariance is PD. However, when $n < d$, the target domain sample covariance is rank-deficient and Assumption 3 is crucial for the identifiablility of $\Delta_\beta$.

**Assumption 4.** The function $G(\cdot)$ is strongly convex with $\gamma > 0$, i.e., for all $w_1, w_2$ in its domain, the following holds:

$$G(w_1) - G(w_2) \ge G'(w_2)(w_1 - w_2) + \gamma \frac{(w_1 - w_2)^2}{2}.$$

Assumption 4 is standard in GLM literature, and it is automatically satisfied when the link function $G'(\cdot) = g(\cdot)$ (7) is linear, i.e., $g(x) = x$ with domain $x \in [0, 1]$. Now, we are ready to present the recovery guarantee for the $\ell_1$ regularized TL for the PS model nuisance parameters.

**Lemma 1** (Transferable guarantee for PS model). Under Assumptions 1, 2, 3 and 4, when the PS model (7) is correctly specified and the difference $\Delta_\beta$ (8) is $s$-sparse, the following holds for the estimator $\widehat{\beta}_\mathrm{t}$ with regularization strength parameter $\lambda_\mathrm{PS} > 0$:

$$\mathbb{P}\left(\left\|\widehat{\beta}_\mathrm{t} - \beta_\mathrm{t}\right\|_1 \ge \frac{5\lambda_\mathrm{PS}}{\gamma}\left(\frac{1}{8\psi^2} + \frac{1}{\psi} + \frac{s}{\phi^2}\right)\right) \le \\ 2d \exp\left(-\frac{2\lambda_\mathrm{PS}^2 n}{125 M_X^2}\right) + 2d \exp\left(-\frac{2\lambda_\mathrm{PS}^2 n_\mathrm{s}}{5d^2 M_X^2}\right). \tag{19}$$

**Remark 1.** As one will see later in next subsection, the above error bound is invoked when we upper bound the error for the estimated propensity scores, i.e., $|g(\boldsymbol{x}_i^\mathrm{T} \beta_\mathrm{t}) - g(\boldsymbol{x}_i^\mathrm{T} \widehat{\beta}_\mathrm{t})|$, which involves applying Hölder's inequality to get

$$\boldsymbol{x}_i^\mathrm{T}(\beta_\mathrm{t} - \widehat{\beta}_\mathrm{t}) \le |\boldsymbol{x}_i^\mathrm{T}(\beta_\mathrm{t} - \widehat{\beta}_\mathrm{t})| \le \|\boldsymbol{x}_i\|_{p_1} \|\beta_\mathrm{t} - \widehat{\beta}_\mathrm{t}\|_{p_2},$$

with $1/p_1 + 1/p_2 = 1$, $p_1, p_2 \geq 1$. Notice that common choices include $(p_1, p_2) = (2, 2)$ and $(\infty, 1)$. As mentioned earlier, we can invoke Restricted Eigenvalue Condition (CANDES & TAO, 2007; Bickel et al., 2009; Meinshausen & Yu, 2009; van de Geer & Bühlmann, 2009) to upper bound $\|\beta_{\mathrm{t}} - \widehat{\beta}_{\mathrm{t}}\|_2$, which scales as $\sqrt{s}$ instead of $s$; however $\|\boldsymbol{x}_i\|_2$ will scale as $\sqrt{n}$ under Assumption 1, which typically dominates the sparsity term in our regime (13). Therefore, the overall error upper bound on ACE estimate will deteriorate to $\mathcal{O}(\sqrt{sn \log d/n})$, compared with $\mathcal{O}(s\sqrt{\log d/n})$ (to be presented below). This explains why we use Compatibility Condition to obtain the $\ell_1$ error bound for the estimated nuisance parameters instead of using Restricted Eigenvalue Condition to get the $\ell_2$ error bound.

### C.2. Guarantee for plug-in TLIPW estimator

To bound the absolute estimation error $|\widehat{\tau}_{\mathrm{TLIPW}} - \tau|$, we additionally need some (mild) technical assumptions:

**Assumption 5.** The target domain outcomes are uniformly bounded, i.e., there exists $M_Y > 0$ such that $|y_i| \leq M_Y, i = 1, \ldots, n$.

This technical assumption helps simplify the analysis; however, our following theoretical analysis will also hold for sub-Gaussian (see Definition 3) outcome random variables as shown by the techniques used in the proof of Theorem 3, case (I).

**Assumption 6.** The propensity scores evaluated on the target domain data are bounded away from zero and one, i.e., there exists $0 < m_g < 1/2$ such that

$$m_g \leq e(\boldsymbol{x}_i) = g(\boldsymbol{x}_i^{\mathrm{T}}\beta_{\mathrm{t}}) \leq 1 - m_g, \quad i = 1, \ldots, n.$$

Assumption 6 is standard for proving the theoretical guarantee of IPW estimator, see Wooldridge et al. (2002); Wooldridge (2007) for classic asymptotic analysis for the IPW estimator's $\sqrt{n}$-consistency and asymptotic normality (cf. Theorems 3.1 and 4.1 (Wooldridge et al., 2002) respectively). Now, by leveraging Hoeffding's inequality, we can establish the following concentration result:

**Lemma 2.** Under Assumptions 5 and 6, for any $t > 0$, we have:

$$\mathbb{P}\left(\left|\frac{1}{n}\sum_{i=1}^{n}\frac{z_i y_i}{g(\boldsymbol{x}_i^{\mathrm{T}}\beta_{\mathrm{t}})} - \frac{(1-z_i)y_i}{1-g(\boldsymbol{x}_i^{\mathrm{T}}\beta_{\mathrm{t}})} - \tau\right| \geq t\right) \leq 4\exp\left(-\frac{m_g^2 t^2 n}{8M_Y^2}\right). \tag{20}$$

Before presenting the non-asymptotic guarantee for TLIPW estimator, we additionally impose the following technical assumption for simplicity:

**Assumption 7.** The link function $g(\cdot)$ is $L$-Lipschitz with constant $L > 0$, i.e., for $x_1, x_2$ in its domain we have $|g(x_1) - g(x_2)| \leq L|x_1 - x_2|$.

Finally, with the help of the above lemmas, we can establish the non-asymptotic upper bound on the absolute estimation error of $\widehat{\tau}_{\mathrm{TLIPW}}$ as follows:

**Theorem 1** (Non-asymptotic recovery guarantee for $\widehat{\tau}_{\mathrm{TLIPW}}$ (12)). Under Assumptions 1, 2, 3, 4, 5, 6 and 7, for any constant $\delta > 0$, if the PS model (7) is correctly specified and the difference $\Delta_\beta$ (8) is $s$-sparse, as $n, n_{\mathrm{s}} \to \infty$, suppose (13) holds, i.e.,

$$s\sqrt{\frac{\log d}{n}} = o(1), \quad d\sqrt{\frac{n}{n_{\mathrm{s}}}} = \mathcal{O}(1),$$

we take $\ell_1$ regularization strength parameter to be

$$\lambda_{\mathrm{PS}} = \sqrt{\frac{5M_X^2 \log(6nd)}{2n}\max\left\{25, \frac{nd^2}{n_{\mathrm{s}}}\right\}}, \tag{21}$$

and we will have

$$\mathbb{P}\left(|\widehat{\tau}_{\mathrm{TLIPW}} - \tau| \leq (1+\delta)\left(C_1 s\sqrt{\frac{\log n + \log d}{n}\max\left\{1, \frac{nd^2}{25n_{\mathrm{s}}}\right\}} + \frac{2M_Y}{m_g}\sqrt{\frac{\log n}{n}}\right)\right) \tag{22}$$

$$\geq 1 - \frac{1}{n},$$

where constant $C_1 = C_1(M_X, M_Y, \psi, \phi; \gamma, m_g, L)$ is defined as:

$$C_1 = \frac{100\sqrt{5}M_X^2 M_Y L}{\sqrt{2}m_g^2 \gamma} \left( \frac{1}{8\psi^2} + \frac{1}{\psi} + \frac{1}{\phi^2} \right).$$

## C.3. Proofs

*Proof outline of Lemma 1.* This proof mostly follows the proof of Theorem 6 in Bastani (2021). The differences in our setting come from: *(i)* The Bernoulli r.v.s are sub-Gaussian with variance bounded by $1/4$, which implies

$$\mathbb{E}[Z - g(\mathbf{X}^\mathrm{T}\beta_\mathrm{t})] = 0, \quad \mathrm{Var}(Z - g(\mathbf{X}^\mathrm{T}\beta_\mathrm{t})) \le 1/4 + 1 = 5/4.$$

We need to substitute the variance terms with this upper bound (i.e., $5/4$).

*(ii)* By Assumption 1, we have

$$\sum_{i=1}^{n} (\mathbf{x}_i)_j^2 \le n M_X^2,$$

where $(\mathbf{x}_i)_j$ denotes the $j$-th element in the vector $\mathbf{x}_i$. This implies that $\sum_{i=1}^{n} (z_i - g(\mathbf{x}_i^\mathrm{T}\beta_\mathrm{t}))(\mathbf{x}_i)_j$ is $(\sqrt{5n}M_X/2)$-sub-Gaussian (cf. Lemma 16 in Bastani (2021)). Notice that this is different from "$\sum_{i=1}^{n}(\mathbf{x}_i)_j^2 = n$" due to the "normalized feature assumption" in the proof of Lemma 4 Bastani (2021). Therefore, in addition to substituting the variance terms as mentioned in (i), we need to include the additional $M_X$ term due to different model assumptions. Lastly, we perform the same modification to Lemma 5 and its proof in Bastani (2021), and these lead to (19). For complete details of the proof, we refer readers to Appendix C in Bastani (2021). $\square$

*Proof of Lemma 2.* For correctly specified propensity score model (7), the IPW estimator is unbiased as shown in eq. (16). Notice that Assumptions 5 and 6 ensures

$$\left| \frac{z_i y_i}{g(\mathbf{x}_i^\mathrm{T}\beta_\mathrm{t})} \right| \le \frac{M_Y}{m_g}.$$

By Hoeffding's inequality, we have

$$\mathbb{P}\left( \left| \frac{1}{n}\sum_{i=1}^{n} \frac{z_i y_i}{g(\mathbf{x}_i^\mathrm{T}\beta_\mathrm{t})} - \mathbb{E}[Y_1] \right| \ge t \right) \le 2\exp\left( -\frac{m_g^2 t^2 n}{2M_Y^2} \right).$$

Similarly, we have $\mathbb{E}\left[ \frac{(1-Z)Y}{1-e(\mathbf{X})} \right] = E[Y_0]$, and we can show

$$\mathbb{P}\left( \left| \frac{1}{n}\sum_{i=1}^{n} \frac{(1-z_i) y_i}{1 - g(\mathbf{x}_i^\mathrm{T}\beta_\mathrm{t})} - \mathbb{E}[Y_0] \right| \ge t \right) \le 2\exp\left( -\frac{m_g^2 t^2 n}{2M_Y^2} \right).$$

Recall that $\tau = \mathbb{E}[Y_1] - \mathbb{E}[Y_0]$, we have

$$\mathbb{P}\left( \left| \frac{1}{n}\sum_{i=1}^{n} \frac{z_i y_i}{g(\mathbf{x}_i^\mathrm{T}\beta_\mathrm{t})} - \frac{(1-z_i) y_i}{1 - g(\mathbf{x}_i^\mathrm{T}\beta_\mathrm{t})} - \tau \right| \ge t \right)$$

$$\le \mathbb{P}\left( \left| \frac{1}{n}\sum_{i=1}^{n} \frac{z_i y_i}{g(\mathbf{x}_i^\mathrm{T}\beta_\mathrm{t})} - \mathbb{E}[Y_1] \right| \ge t/2 \right) + \mathbb{P}\left( \left| \frac{1}{n}\sum_{i=1}^{n} \frac{(1-z_i) y_i}{1 - g(\mathbf{x}_i^\mathrm{T}\beta_\mathrm{t})} - \mathbb{E}[Y_0] \right| \ge t/2 \right)$$

$$\le 4\exp\left( -\frac{m_g^2 t^2 n}{8M_Y^2} \right).$$

We complete the proof. $\square$

*Proof of Theorem 1.* One one hand, plugging the regularization parameter choice (21) into (19) yields:

$$\mathbb{P}\left(\left\|\widehat{\beta}_{\mathrm{t}} - \beta_{\mathrm{t}}\right\|_1 \geq \frac{5\lambda_{\mathrm{PS}}}{\gamma}\left(\frac{1}{8\psi^2} + \frac{1}{\psi} + \frac{s}{\phi^2}\right)\right) \leq \frac{2}{3n}. \tag{23}$$

On the other hand, by setting $t = \frac{2M_Y}{m_g}\sqrt{\frac{\log(12n)}{n}}$ in eq. (20) we have

$$\mathbb{P}\left(\left|\frac{1}{n}\sum_{i=1}^{n}\frac{z_i y_i}{g(\boldsymbol{x}_i^{\mathrm{T}}\beta_{\mathrm{t}})} - \frac{(1-z_i)y_i}{1-g(\boldsymbol{x}_i^{\mathrm{T}}\beta_{\mathrm{t}})} - \tau\right| \geq \frac{2M_Y}{m_g}\sqrt{\frac{\log(12n)}{n}}\right) \leq \frac{1}{3n}. \tag{24}$$

Due to Assumptions 5 and 6, we have

$$\left|\frac{1}{n}\sum_{i=1}^{n}\frac{z_i y_i}{g(\boldsymbol{x}_i^{\mathrm{T}}\beta_{\mathrm{t}})} - \frac{z_i y_i}{g(\boldsymbol{x}_i^{\mathrm{T}}\widehat{\beta}_{\mathrm{t}})}\right| \leq \frac{1}{n}\sum_{i=1}^{n}\frac{M_Y|g(\boldsymbol{x}_i^{\mathrm{T}}\beta_{\mathrm{t}}) - g(\boldsymbol{x}_i^{\mathrm{T}}\widehat{\beta}_{\mathrm{t}})|}{m_g\left(m_g - |g(\boldsymbol{x}_i^{\mathrm{T}}\beta_{\mathrm{t}}) - g(\boldsymbol{x}_i^{\mathrm{T}}\widehat{\beta}_{\mathrm{t}})|\right)}. \tag{25}$$

Since $g(\cdot)$ is $L$-Lipschitz, we have

$$|g(\boldsymbol{x}_i^{\mathrm{T}}\beta_{\mathrm{t}}) - g(\boldsymbol{x}_i^{\mathrm{T}}\widehat{\beta}_{\mathrm{t}})| \leq L|\boldsymbol{x}_i^{\mathrm{T}}(\beta_{\mathrm{t}} - \widehat{\beta}_{\mathrm{t}})| \leq L\|\boldsymbol{x}_i\|_\infty\left\|\widehat{\beta}_{\mathrm{t}} - \beta_{\mathrm{t}}\right\|_1,$$

where the last inequality comes from the Hölder's inequality. Due to Assumption 5 and the fact that $f(x) = x/(m_g - x)$ monotonically increase on domain $0 \leq x < m_g$, we can further bound the right hand side (RHS) of (25) as follows:

$$\left|\frac{1}{n}\sum_{i=1}^{n}\frac{z_i y_i}{g(\boldsymbol{x}_i^{\mathrm{T}}\beta_{\mathrm{t}})} - \frac{z_i y_i}{g(\boldsymbol{x}_i^{\mathrm{T}}\widehat{\beta}_{\mathrm{t}})}\right| \leq \frac{1}{n}\sum_{i=1}^{n}\frac{M_X M_Y L\left\|\widehat{\beta}_{\mathrm{t}} - \beta_{\mathrm{t}}\right\|_1}{m_g\left(m_g - M_X L\left\|\widehat{\beta}_{\mathrm{t}} - \beta_{\mathrm{t}}\right\|_1\right)}$$
$$\leq \frac{1}{n}\sum_{i=1}^{n}\frac{M_X M_Y L\left\|\widehat{\beta}_{\mathrm{t}} - \beta_{\mathrm{t}}\right\|_1}{m_g^2/2}. \tag{26}$$

The above inequality will hold since, for large enough $n, n_{\mathrm{s}}$ and in the regime (13), Lemma 1 guarantees $M_X L\left\|\widehat{\beta}_{\mathrm{t}} - \beta_{\mathrm{t}}\right\|_1 \to 0$, and therefore we will have $M_X L\left\|\widehat{\beta}_{\mathrm{t}} - \beta_{\mathrm{t}}\right\|_1 \leq m_g/2$. Similarly, we can obtain

$$\left|\frac{1}{n}\sum_{i=1}^{n}\frac{(1-z_i)y_i}{1-g(\boldsymbol{x}_i^{\mathrm{T}}\beta_{\mathrm{t}})} - \frac{(1-z_i)y_i}{1-g(\boldsymbol{x}_i^{\mathrm{T}}\widehat{\beta}_{\mathrm{t}})}\right| \leq \frac{1}{n}\sum_{i=1}^{n}\frac{M_X M_Y L\left\|\widehat{\beta}_{\mathrm{t}} - \beta_{\mathrm{t}}\right\|_1}{m_g^2/2}. \tag{27}$$

Now, (23) and (24) tell us that, with probability as least $1 - 1/n$,

$$|\widehat{\tau}_{\mathrm{TLIPW}} - \tau|$$
$$\leq \left|\widehat{\tau}_{\mathrm{TLIPW}} - \frac{1}{n}\sum_{i=1}^{n}\frac{z_i y_i}{g(\boldsymbol{x}_i^{\mathrm{T}}\beta_{\mathrm{t}})} - \frac{(1-z_i)y_i}{1-g(\boldsymbol{x}_i^{\mathrm{T}}\beta_{\mathrm{t}})}\right| + \left|\frac{1}{n}\sum_{i=1}^{n}\frac{z_i y_i}{g(\boldsymbol{x}_i^{\mathrm{T}}\beta_{\mathrm{t}})} - \frac{(1-z_i)y_i}{1-g(\boldsymbol{x}_i^{\mathrm{T}}\beta_{\mathrm{t}})} - \tau\right|$$
$$< \frac{20M_X M_Y L\lambda_{\mathrm{PS}}}{m_g^2\gamma}\left(\frac{1}{8\psi^2} + \frac{1}{\psi} + \frac{s}{\phi^2}\right) + \frac{2M_Y}{m_g}\sqrt{\frac{\log(12n)}{n}}$$
$$\leq \frac{20M_X M_Y L\lambda_{\mathrm{PS}}}{m_g^2\gamma}\left(\frac{1}{8\psi^2} + \frac{1}{\psi} + \frac{1}{\phi^2}\right)s + \frac{2M_Y}{m_g}\sqrt{\frac{\log(12n)}{n}}.$$

Plugging the $\lambda_{\mathrm{PS}}$ choice (21) into the above equation, and notice that, for any constant $\delta > 0$, for large enough $n$ the following holds:

$$\sqrt{\log(12n)} = \sqrt{\log 12 + \log n} \leq (1+\delta)\sqrt{\log n}, \quad \sqrt{\log(6nd)} \leq (1+\delta)\sqrt{\log(nd)}.$$

We can obtain the non-asymptotic result in eq. (22). Now we complete the proof.

$\square$

# D. Non-Asymptotic Recovery Guarantee for TLOR estimator

## D.1. Guarantee for OR model nuisance parameter estimation with knowledge transfer

Now we prove the non-asymptotic guarantee for our proposed TLOR estimator.

**Definition 3.** A random variable $Z \in \mathbb{R}$ is $\sigma$-sub-Gaussian if $\mathbb{E}\left[e^{tz}\right] \leq e^{\sigma^2 t^2/2}$ for all $t \in \mathbb{R}$.

Many classical distributions are subgaussian; typical examples include any bounded, centered distribution, or the normal distribution. For $z \in \{0, 1\}$, we denote the "noise terms" in the OR models (10) as follows:

$$\epsilon_z = Y_z - \mathbb{E}[Y_z|\boldsymbol{X}] = Y_z - \boldsymbol{X}^{\mathrm{T}}\alpha_{z,\mathrm{t}},$$
$$\epsilon_{z,\mathrm{s}} = Y_{z,\mathrm{s}} - \mathbb{E}[Y_{z,\mathrm{s}}|\boldsymbol{X}_\mathrm{s}] = Y_{z,\mathrm{s}} - \boldsymbol{X}_\mathrm{s}^{\mathrm{T}}\alpha_{z,\mathrm{s}}.$$

**Assumption 8.** The noise terms are sub-Gaussian, i.e., for $z \in \{0, 1\}$, there exist constants $\sigma, \sigma_\mathrm{s} > 0$ such that $\epsilon_z$ is $\sigma$-sub-Gaussian, and $\epsilon_{z,\mathrm{s}}$ is $\sigma_\mathrm{s}$-sub-Gaussian.

**Assumption 9.** For $z \in \{0, 1\}$, the index set $\mathcal{I} = \mathrm{supp}(\Delta_{\alpha,z})$ (11) and target domain sample covariance matrix $\Sigma$ (18) meet the compatibility condition with $\phi_z > 0$.

**Lemma 3** (Transferable guarantee for OR model, cf. Theorem 5 (Bastani, 2021)). Under Assumptions 1, 2, (8) and 9, assume the sample balanceness condition (14) holds, for $z \in \{0, 1\}$, when the OR model (10) is correctly specified and the difference $\Delta_{\alpha,z}$ (11) is $s_z$-sparse, i.e.,

$$\|\Delta_{\alpha,0}\|_0 \leq s_0, \quad \|\Delta_{\alpha,1}\|_0 \leq s_1,$$

the following holds for the estimator $\widehat{\alpha}_{z,\mathrm{t}}$ with regularization strength parameter $\lambda_{\mathrm{OR}} > 0$:

$$
\mathbb{P}\left(\|\widehat{\alpha}_{z,\mathrm{t}} - \alpha_{z,\mathrm{t}}\|_1 \geq 5\lambda_{\mathrm{OR}}\left(\frac{1}{4\psi^2} + \frac{1}{\psi} + \frac{s_z}{2\phi_z^2}\right)\right) \leq
$$
$$
2d\exp\left(-\frac{rn\lambda_{\mathrm{OR}}^2}{200\sigma^2 M_X^2}\right) + 2d\exp\left(-\frac{rn_\mathrm{s}\lambda_{\mathrm{OR}}^2}{2d^2\sigma_\mathrm{s}^2 M_X^2}\right). \tag{28}
$$

## D.2. Guarantee for plug-in TLOR estimator

Here, we impose an additional technical assumption that the covariates in the target domain are sub-Gaussian such that there exists constant $\sigma_X > 0$:

$$
\mathbb{P}\left(\left|\frac{1}{n}\sum_{i=1}^n \boldsymbol{x}_i^{\mathrm{T}}\alpha_{z,\mathrm{t}} - \mathbb{E}[Y_z]\right| > t\right) \leq 2\exp\left(-\frac{nt^2}{2\sigma_X^2}\right), \quad z \in \{0, 1\}. \tag{29}
$$

**Theorem 2** (Non-asymptotic recovery guarantee for $\widehat{\tau}_{\mathrm{TLDR}}$ (12)). Under Assumptions 1, 2, 8 and 9, suppose the sample balanceness condition (14) holds and the covariates in target domain follow a sub-Gaussian distribution (29), as $n, n_\mathrm{s} \to \infty$, suppose (13) holds, for any constant $\delta > 0$, when the OR model (10) is correctly specified with $s_z$-sparse $\Delta_{z,\alpha}$ (11) (for $z \in \{0, 1\}$), i.e.,

$$\|\Delta_{\alpha,0}\|_0 \leq s_0, \quad \|\Delta_{\alpha,1}\|_0 \leq s_1,$$

then by taking

$$
\lambda_{\mathrm{OR}} = \sqrt{2M_X^2 \log(12nd) \max\left\{\frac{100\sigma^2}{rn}, \frac{d^2\sigma_\mathrm{s}^2}{rn_\mathrm{s}}\right\}}, \tag{30}
$$

we will have

$$
\mathbb{P}\left(|\widehat{\tau}_{\mathrm{TLOR}} - \tau| \leq (1+\delta)\left(C_2\sigma(s_0 + s_1)\sqrt{\frac{\log n + \log d}{rn}}\max\left\{100, \frac{n\sigma_\mathrm{s}^2 d^2}{n_\mathrm{s}\sigma^2}\right\}\right.\right.
$$
$$
\left.\left.+ 2\sigma_X\sqrt{\frac{2\log n}{rn}}\right)\right) \geq 1 - \frac{1}{n}. \tag{31}
$$

where constants $C_2 = C_2(M_X, \psi, \phi_0, \phi_1; m_g)$ and $C_3 = C_3(m_g)$ are defined as follows:

$$
C_2 = 5\sqrt{2}M_X^2\left(\frac{1}{m_g} - 1\right)\left(\frac{1}{2\psi^2} + \frac{2}{\psi} + \frac{1}{2\phi_0^2} + \frac{1}{2\phi_1^2}\right), \quad C_3 = 2\sqrt{\frac{2}{m_g}}. \tag{32}
$$

## D.3. Proof

The proof of Lemma 3 mostly follows that of Theorem 5 in Bastani (2021) and we omit it here. We only give detailed proof of the main theorem.

*Proof of Theorem 2.* The absolute error of the TLOR estimator (12) can be decomposed as follows:

$$|\widehat{\tau}_{\text{TLOR}} - \tau| \leq \left| \frac{1}{n_1} \sum_{z_i=1} \boldsymbol{x}_i^{\text{T}} (\widehat{\alpha}_{1,\text{t}} - \alpha_{1,\text{t}}) \right| + \left| \frac{1}{n_0} \sum_{z_i=0} \boldsymbol{x}_i^{\text{T}} (\widehat{\alpha}_{0,\text{t}} - \alpha_{0,\text{t}}) \right|$$

$$+ \left| \frac{1}{n_1} \sum_{z_i=1} \boldsymbol{x}_i^{\text{T}} \alpha_{1,\text{t}} - \frac{1}{n_0} \sum_{z_i=0} \boldsymbol{x}_i^{\text{T}} \alpha_{0,\text{t}} - \tau \right|$$

$$\leq M_X \left( \|\widehat{\alpha}_{1,\text{t}} - \alpha_{1,\text{t}}\|_1 + \|\widehat{\alpha}_{0,\text{t}} - \alpha_{0,\text{t}}\|_1 \right)$$

$$+ \left| \frac{1}{n_1} \sum_{z_i=1} \boldsymbol{x}_i^{\text{T}} \alpha_{1,\text{t}} - \frac{1}{n_0} \sum_{z_i=0} \boldsymbol{x}_i^{\text{T}} \alpha_{0,\text{t}} - \tau \right|.$$

On one hand, by setting the RHS of (29) as $1/(6n)$, we have that, with probability at least $1 - 1/(3n)$, the following holds:

$$\left| \frac{1}{n} \sum_{i=1}^{n} \boldsymbol{x}_i^{\text{T}} \alpha_{1,\text{t}} - \boldsymbol{x}_i^{\text{T}} \alpha_{0,\text{t}} - \tau \right| \leq 2 \sqrt{\frac{2\sigma_X^2 \log(12n)}{n}}.$$

One the other hand, following Lemma 3 and taking $\lambda_{\text{OR}}$ as in eq. (30), we will have that, with probability at least $1 - 2/(3n)$,

$$\|\widehat{\alpha}_{1,\text{t}} - \alpha_{1,\text{t}}\|_1 + \|\widehat{\alpha}_{0,\text{t}} - \alpha_{0,\text{t}}\|_1 \leq 5\lambda_{\text{OR}} \left( \frac{1}{2\psi^2} + \frac{2}{\psi} + \frac{s_0}{2\phi_0^2} + \frac{s_1}{2\phi_1^2} \right).$$

Now, combing the above inequalities, we will have that, with probability at least $1 - 1/n$, the following holds:

$$|\widehat{\tau}_{\text{TLOR}} - \tau| \leq 2\sqrt{\frac{2\sigma_X^2 \log(12n)}{n}} + 5M_X \left( \frac{1}{m_g} - 1 \right) \lambda_{\text{OR}} \left( \frac{1}{2\psi^2} + \frac{2}{\psi} + \frac{s_0}{2\phi_0^2} + \frac{s_1}{2\phi_1^2} \right)$$

$$= \mathcal{O} \left( (s_0 + s_1) \sqrt{\log d} \left( \frac{1}{\sqrt{rn}} + \frac{d}{\sqrt{rn_{\text{s}}}} \right) \right).$$

To get (31), we only need to notice that, for any constant $\delta > 0$, for large enough $n$ the following holds:

$$\sqrt{\log(12n)} = \sqrt{\log 12 + \log n} \leq (1+\delta)\sqrt{\log n}.$$

We can handle the $\log(12nd)$ term in $\lambda_{\text{OR}}$ (30) in a similar manner and obtain $\sqrt{\log(12nd)} \leq (1+\delta)\sqrt{\log(nd)}$. Now, we obtain the non-asymptotic result in eq. (31) and complete the proof. $\square$

## E. Non-Asymptotic Recovery Guarantee for TLDR estimator

### E.1. Guarantee for plug-in TLDR estimator

We impose another technical assumption as Bastani (2021) did (see Assumption 1 therein):

**Assumption 10.** The ground truth target domain OR model parameters are bounded, i.e., for $z \in \{0, 1\}$, there exists $M_\alpha > 0$ such that $\|\alpha_{z,\text{t}}\|_1 < M_\alpha$.

Similarly, due to the doubly-robustness of the DR estimator, we have the following non-asymptotic recovery guarantee:

**Theorem 3** (Non-asymptotic recovery guarantee for $\widehat{\tau}_{\text{TLDR}}$ (12))**.** Under Assumptions 1 and 2, as $n, n_{\text{s}} \to \infty$, suppose (13) holds. For any constant $\delta > 0$:

(I) When the PS model (7) is correctly specified and $\Delta_\beta$ (8) is $s$-sparse, if we additionally assume Assumptions 3, 4, 5, 6, 7 and 10 hold, then by taking

$$\lambda_{\mathrm{PS}} = \sqrt{\frac{5M_X^2 \log(10nd)}{2n} \max\left\{25, \frac{nd^2}{n_{\mathrm{s}}}\right\}}, \tag{33}$$

we will have

$$\mathbb{P}\left(|\widehat{\tau}_{\mathrm{TLDR}} - \tau| \le (1+\delta)\left(C_4 s\sqrt{\frac{\log n + \log d}{n} \max\left\{1, \frac{25nd^2}{n_{\mathrm{s}}}\right\}} + C_5\sqrt{\frac{\log n}{n}}\right)\right) \ge 1 - \frac{1}{n}. \tag{34}$$

where constants $C_4 = C_4(M_X, M_Y, \psi, \phi; \gamma, m_g, L; M_\alpha)$ and $C_5 = C_5(M_X, M_Y; m_g; M_\alpha)$ are defined as follows:

$$C_4 = \frac{100\sqrt{5}M_X^2 (M_X M_\alpha + M_Y/m_g) L}{\sqrt{2}m_g\gamma}\left(\frac{1}{8\psi^2} + \frac{1}{\psi} + \frac{1}{\phi^2}\right), \quad C_5 = 2\frac{M_Y + \sqrt{10}M_X M_\alpha}{m_g}.$$

(II) When the OR model (10) is correctly specified and $\Delta_{z,\alpha}$ (11) is $s_z$-sparse (for $z \in \{0,1\}$), i.e.,

$$\|\Delta_{\alpha,0}\|_0 \le s_0, \quad \|\Delta_{\alpha,1}\|_0 \le s_1,$$

under Assumptions 8, 9, and we further assume the sample balanceness condition (14) holds and the covariates in the target domain are sub-Gaussian (29), we strengthen Assumption 6 by assuming the link function $g(\cdot)$ (7) takes value on $[m_g, 1 - m_g]$, then by taking

$$\lambda_{\mathrm{OR}} = \sqrt{2M_X^2 \log(16nd) \max\left\{\frac{100\sigma^2}{rn}, \frac{d^2\sigma_{\mathrm{s}}^2}{rn_{\mathrm{s}}}\right\}}, \tag{35}$$

we will have

$$\mathbb{P}\left(|\widehat{\tau}_{\mathrm{TLDR}} - \tau| \le (1+\delta)\left(C_2\sigma(s_0 + s_1)\sqrt{\frac{\log n + \log d}{rn} \max\left\{100, \frac{n\sigma_{\mathrm{s}}^2 d^2}{n_{\mathrm{s}}\sigma^2}\right\}} \right.\right.$$
$$\left.\left. + C_3\sigma\sqrt{\frac{\log n}{n}} + 2\sigma_X\sqrt{\frac{2\log n}{rn}}\right)\right) \ge 1 - \frac{1}{n}. \tag{36}$$

where constants $C_2 = C_2(M_X, \psi, \phi_0, \phi_1; m_g)$ and $C_3 = C_3(m_g)$ are defined in eq. (32).

### E.2. Proof

*Proof of Theorem 3.* We consider two cases: (i) the PS model is correctly specified and (ii) the OR model is correctly specified.

**Case (I): the PS model is correctly specified.** This proof closely resembles that of Theorem 1. We only need to showed the augmented terms in the DR estimator, i.e.,

$$\frac{1}{n}\sum_{i=1}^n \frac{\boldsymbol{x}_i^{\mathrm{T}}\widehat{\alpha}_{1,\mathrm{t}}(z_i - g(\boldsymbol{x}_i^{\mathrm{T}}\widehat{\beta}_{\mathrm{t}}))}{g(\boldsymbol{x}_i^{\mathrm{T}}\widehat{\beta}_{\mathrm{t}})}, \quad \frac{1}{n}\sum_{i=1}^n \frac{\boldsymbol{x}_i^{\mathrm{T}}\widehat{\alpha}_{0,\mathrm{t}}(z_i - g(\boldsymbol{x}_i^{\mathrm{T}}\widehat{\beta}_{\mathrm{t}}))}{1 - g(\boldsymbol{x}_i^{\mathrm{T}}\widehat{\beta}_{\mathrm{t}})},$$

are very close to zero with high probability, which is straightforward since they both have zero mean under the correct PS model specification assumption.

To simplify our proof below, we impose Assumption 10, which implies that, when $n, n_{\mathrm{s}} \to \infty$, in regime (13), we will have $\|\widehat{\alpha}_{z,\mathrm{t}}\|_1 \le M_\alpha$ due to Lemma 3. Similarly, Assumption 6 ensures that $m_g/2 \le g(\boldsymbol{x}_i^{\mathrm{T}}\widehat{\beta}_{\mathrm{t}}) \le 1 - m_g/2$ for sufficiently large $n, n_{\mathrm{s}}$. Now, we can show that

$$\sum_{i=1}^n \frac{\boldsymbol{x}_i^{\mathrm{T}}\widehat{\alpha}_{1,\mathrm{t}}(z_i - g(\boldsymbol{x}_i^{\mathrm{T}}\beta_{\mathrm{t}}))}{g(\boldsymbol{x}_i^{\mathrm{T}}\widehat{\beta}_{\mathrm{t}})}$$

is $(\sqrt{5n}M_X M_\alpha/m_g)$-sub-Gaussian. Notice that Bernoulli r.v. has variance bounded by $1/4$ and

$$\left|\frac{\boldsymbol{x}_i^\mathsf{T}\widehat{\alpha}_{1,\mathrm{t}}}{g(\boldsymbol{x}_i^\mathsf{T}\widehat{\beta}_{\mathrm{t}})}\right| \leq \frac{2M_X M_\alpha}{m_g}.$$

Thus, we have

$$\mathbb{P}\left(\left|\frac{1}{n}\sum_{i=1}^{n}\frac{\boldsymbol{x}_i^\mathsf{T}\widehat{\alpha}_{1,\mathrm{t}}(z_i - g(\boldsymbol{x}_i^\mathsf{T}\beta_{\mathrm{t}}))}{g(\boldsymbol{x}_i^\mathsf{T}\widehat{\beta}_{\mathrm{t}})}\right| > t\right) \leq 2\exp\left(-\frac{nm_g^2 t^2}{10 M_X^2 M_\alpha^2}\right). \tag{37}$$

Similarly, we can show that

$$\mathbb{P}\left(\left|\frac{1}{n}\sum_{i=1}^{n}\frac{\boldsymbol{x}_i^\mathsf{T}\widehat{\alpha}_{0,\mathrm{t}}(z_i - g(\boldsymbol{x}_i^\mathsf{T}\beta_{\mathrm{t}}))}{1 - g(\boldsymbol{x}_i^\mathsf{T}\widehat{\beta}_{\mathrm{t}})}\right| > t\right) \leq 2\exp\left(-\frac{nm_g^2 t^2}{10 M_X^2 M_\alpha^2}\right). \tag{38}$$

We take RHS of Equations 37 and 38 to be $1/(5n)$, and therefore with probability at least $1 - 2/(5n)$ we have:

$$\left|-\frac{1}{n}\sum_{i=1}^{n}\frac{\boldsymbol{x}_i^\mathsf{T}\widehat{\alpha}_{1,\mathrm{t}}(z_i - g(\boldsymbol{x}_i^\mathsf{T}\widehat{\beta}_{\mathrm{t}}))}{g(\boldsymbol{x}_i^\mathsf{T}\widehat{\beta}_{\mathrm{t}})} + \frac{1}{n}\sum_{i=1}^{n}\frac{\boldsymbol{x}_i^\mathsf{T}\widehat{\alpha}_{0,\mathrm{t}}(z_i - g(\boldsymbol{x}_i^\mathsf{T}\widehat{\beta}_{\mathrm{t}}))}{1 - g(\boldsymbol{x}_i^\mathsf{T}\widehat{\beta}_{\mathrm{t}})}\right|$$
$$\leq \frac{2M_X M_\alpha \sqrt{10\log(10n)}}{m_g\sqrt{n}} + \frac{4LM_X^2 M_\alpha \|\beta_{\mathrm{t}} - \widehat{\beta}_{\mathrm{t}}\|_1}{m_g}. \tag{39}$$

By taking $\lambda_{\mathrm{PS}}$ as in eq. (33) and following the proof of Theorem 1, we have that with probability at least $1 - 1/n$:

$$\begin{aligned}
&|\widehat{\tau}_{\mathrm{TLDR}} - \tau| \\
&\leq \left|\widehat{\tau}_{\mathrm{TLIPW}} - \frac{1}{n}\sum_{i=1}^{n}\frac{z_i y_i}{g(\boldsymbol{x}_i^\mathsf{T}\beta_{\mathrm{t}})} - \frac{(1 - z_i)y_i}{1 - g(\boldsymbol{x}_i^\mathsf{T}\beta_{\mathrm{t}})}\right| \\
&+ \left|\frac{1}{n}\sum_{i=1}^{n}\frac{z_i y_i}{g(\boldsymbol{x}_i^\mathsf{T}\beta_{\mathrm{t}})} - \frac{(1 - z_i)y_i}{1 - g(\boldsymbol{x}_i^\mathsf{T}\beta_{\mathrm{t}})} - \tau\right| \\
&+ \left|-\frac{1}{n}\sum_{i=1}^{n}\frac{\boldsymbol{x}_i^\mathsf{T}\widehat{\alpha}_{1,\mathrm{t}}(z_i - g(\boldsymbol{x}_i^\mathsf{T}\widehat{\beta}_{\mathrm{t}}))}{g(\boldsymbol{x}_i^\mathsf{T}\widehat{\beta}_{\mathrm{t}})} + \frac{1}{n}\sum_{i=1}^{n}\frac{\boldsymbol{x}_i^\mathsf{T}\widehat{\alpha}_{0,\mathrm{t}}(z_i - g(\boldsymbol{x}_i^\mathsf{T}\widehat{\beta}_{\mathrm{t}}))}{1 - g(\boldsymbol{x}_i^\mathsf{T}\widehat{\beta}_{\mathrm{t}})}\right| \\
&\leq \frac{20M_X\left(M_\alpha M_X + \frac{M_Y}{m_g}\right)L\lambda_{\mathrm{PS}}}{m_g\gamma}\left(\frac{1}{8\psi^2} + \frac{1}{\psi} + \frac{s}{\phi^2}\right) + \frac{2M_Y\sqrt{\log(20n)} + 2M_X M_\alpha\sqrt{10\log(10n)}}{m_g\sqrt{n}} \\
&= \mathcal{O}\left(s\sqrt{\log d}\left(\frac{1}{\sqrt{n}} + \frac{d}{\sqrt{n_\mathrm{s}}}\right)\right).
\end{aligned}$$

To get (34), we only need to notice that, for any constant $\delta > 0$, for large enough $n$ the following holds:

$$\sqrt{\log(10n)} < \sqrt{\log(20n)} = \sqrt{\log 20 + \log n} \leq (1 + \delta)\sqrt{\log n}.$$

We can handle the $\log(10nd)$ term in $\lambda_{\mathrm{PS}}$ (33) in a similar manner and obtain $\sqrt{\log(10nd)} \leq (1 + \delta)\sqrt{\log(nd)}$. Now, we obtain the non-asymptotic result in eq. (34) and complete the proof.

**Case (II): the OR model is correctly specified.** We rewrite our TLDR estimator (12) as follows:

$$\widehat{\tau}_{\mathrm{TLDR}} = \frac{1}{n}\sum_{i=1}^{n}\boldsymbol{x}_i^\mathsf{T}\widehat{\alpha}_{1,\mathrm{t}} + \frac{z_i(y_i - \boldsymbol{x}_i^\mathsf{T}\widehat{\alpha}_{1,\mathrm{t}})}{g(\boldsymbol{x}_i^\mathsf{T}\widehat{\beta}_{\mathrm{t}})} - \boldsymbol{x}_i^\mathsf{T}\widehat{\alpha}_{0,\mathrm{t}} - \frac{(1 - z_i)(y_i - \boldsymbol{x}_i^\mathsf{T}\widehat{\alpha}_{0,\mathrm{t}})}{1 - g(\boldsymbol{x}_i^\mathsf{T}\widehat{\beta}_{\mathrm{t}})}.$$

We decompose the estimator error as follows:

$$|\widehat{\tau}_{\text{TLDR}} - \tau| \leq \left| \frac{1}{n} \sum_{i=1}^{n} \boldsymbol{x}_i^{\text{T}} \alpha_{1,\text{t}} - \boldsymbol{x}_i^{\text{T}} \alpha_{0,\text{t}} - \tau \right| + \left| \frac{1}{n} \sum_{i=1}^{n} \left( 1 - \frac{z_i}{g(\boldsymbol{x}_i^{\text{T}} \widehat{\beta}_{\text{t}})} \right) \boldsymbol{x}_i^{\text{T}} (\widehat{\alpha}_{1,\text{t}} - \alpha_{1,\text{t}}) \right|$$

$$+ \left| \frac{1}{n} \sum_{i=1}^{n} \left( 1 - \frac{1 - z_i}{1 - g(\boldsymbol{x}_i^{\text{T}} \widehat{\beta}_{\text{t}})} \right) \boldsymbol{x}_i^{\text{T}} (\widehat{\alpha}_{0,\text{t}} - \alpha_{0,\text{t}}) \right| + \left| \frac{1}{n} \sum_{i=1}^{n} \frac{z_i (y_i - \boldsymbol{x}_i^{\text{T}} \alpha_{1,\text{t}})}{g(\boldsymbol{x}_i^{\text{T}} \widehat{\beta}_{\text{t}})} \right|$$

$$+ \left| \frac{1}{n} \sum_{i=1}^{n} \frac{(1 - z_i)(y_i - \boldsymbol{x}_i^{\text{T}} \alpha_{0,\text{t}})}{1 - g(\boldsymbol{x}_i^{\text{T}} \widehat{\beta}_{\text{t}})} \right|.$$

Notice that we strengthen the Assumption 6 that link function $g(\cdot)$ only takes value on $[m_g, 1 - m_g]$, which gives us

$$\left| \frac{1}{n} \sum_{i=1}^{n} \left( 1 - \frac{z_i}{g(\boldsymbol{x}_i^{\text{T}} \widehat{\beta}_{\text{t}})} \right) \boldsymbol{x}_i^{\text{T}} (\widehat{\alpha}_{1,\text{t}} - \alpha_{1,\text{t}}) \right| \leq \left( \frac{1}{m_g} - 1 \right) \frac{1}{n} \sum_{i=1}^{n} |\boldsymbol{x}_i^{\text{T}} (\widehat{\alpha}_{1,\text{t}} - \alpha_{1,\text{t}})|,$$

$$\left| \frac{1}{n} \sum_{i=1}^{n} \left( 1 - \frac{1 - z_i}{1 - g(\boldsymbol{x}_i^{\text{T}} \widehat{\beta}_{\text{t}})} \right) \boldsymbol{x}_i^{\text{T}} (\widehat{\alpha}_{0,\text{t}} - \alpha_{0,\text{t}}) \right| \leq \left( \frac{1}{m_g} - 1 \right) \frac{1}{n} \sum_{i=1}^{n} |\boldsymbol{x}_i^{\text{T}} (\widehat{\alpha}_{0,\text{t}} - \alpha_{0,\text{t}})|.$$

Combing the above derivations with Assumption 1, we have

$$|\widehat{\tau}_{\text{TLDR}} - \tau| \leq \left| \frac{1}{n} \sum_{i=1}^{n} \boldsymbol{x}_i^{\text{T}} \alpha_{1,\text{t}} - \boldsymbol{x}_i^{\text{T}} \alpha_{0,\text{t}} - \tau \right| + M_X \left( \frac{1}{m_g} - 1 \right) \left( \|\widehat{\alpha}_{1,\text{t}} - \alpha_{1,\text{t}}\|_1 + \|\widehat{\alpha}_{0,\text{t}} - \alpha_{0,\text{t}}\|_1 \right)$$

$$+ \left| \frac{1}{n} \sum_{i=1}^{n} \frac{z_i (y_i - \boldsymbol{x}_i^{\text{T}} \alpha_{1,\text{t}})}{g(\boldsymbol{x}_i^{\text{T}} \widehat{\beta}_{\text{t}})} \right| + \left| \frac{1}{n} \sum_{i=1}^{n} \frac{(1 - z_i)(y_i - \boldsymbol{x}_i^{\text{T}} \alpha_{0,\text{t}})}{1 - g(\boldsymbol{x}_i^{\text{T}} \widehat{\beta}_{\text{t}})} \right|.$$

*(i)* Since the OR model is assumed to be correct, we have $\mathbb{E}[Z(Y - \boldsymbol{X}^{\text{T}} \alpha_{Z,\text{t}})] = 0$. Under Assumption 6, we can show:

$$\mathbb{P} \left( \left| \frac{1}{n} \sum_{i=1}^{n} \frac{z_i (y_i - \boldsymbol{x}_i^{\text{T}} \alpha_{1,\text{t}})}{g(\boldsymbol{x}_i^{\text{T}} \widehat{\beta}_{\text{t}})} \right| > t \right)$$

$$= \mathbb{P} \left( \left| \frac{1}{n} \sum_{z_i = 1} \frac{y_i - \boldsymbol{x}_i^{\text{T}} \alpha_{1,\text{t}}}{g(\boldsymbol{x}_i^{\text{T}} \widehat{\beta}_{\text{t}})} \right| > t \right) \leq 2 \exp \left( -\frac{r n m_g^2 t^2}{2 \sigma^2} \right). \tag{40}$$

Similarly we will get:

$$\mathbb{P} \left( \left| \frac{1}{n} \sum_{i=1}^{n} \frac{(1 - z_i)(y_i - \boldsymbol{x}_i^{\text{T}} \alpha_{0,\text{t}})}{1 - g(\boldsymbol{x}_i^{\text{T}} \widehat{\beta}_{\text{t}})} \right| > t \right)$$

$$= \mathbb{P} \left( \left| \frac{1}{n} \sum_{z_i = 0} \frac{y_i - \boldsymbol{x}_i^{\text{T}} \alpha_{0,\text{t}}}{1 - g(\boldsymbol{x}_i^{\text{T}} \widehat{\beta}_{\text{t}})} \right| > t \right) \leq 2 \exp \left( -\frac{r n m_g^2 t^2}{2 \sigma^2} \right). \tag{41}$$

By setting the RHS of the above two inequalities as $1/(8n)$, we have that, with probability at least $1 - 1/(4n)$, the following holds:

$$\left| \frac{1}{n} \sum_{i=1}^{n} \frac{z_i (y_i - \boldsymbol{x}_i^{\text{T}} \alpha_{1,\text{t}})}{g(\boldsymbol{x}_i^{\text{T}} \widehat{\beta}_{\text{t}})} \right| + \left| \frac{1}{n} \sum_{i=1}^{n} \frac{(1 - z_i)(y_i - \boldsymbol{x}_i^{\text{T}} \alpha_{0,\text{t}})}{1 - g(\boldsymbol{x}_i^{\text{T}} \widehat{\beta}_{\text{t}})} \right| \leq 2 \sqrt{\frac{2 \sigma^2 \log(16n)}{r n m_g^2}}.$$

*(ii)* Similar to the proof of Theorem 2, by setting the RHS of (29) as $1/(8n)$, we have that, with probability at least $1 - 1/(4n)$, the following holds:

$$\left| \frac{1}{n} \sum_{i=1}^{n} \boldsymbol{x}_i^{\text{T}} \alpha_{1,\text{t}} - \boldsymbol{x}_i^{\text{T}} \alpha_{0,\text{t}} - \tau \right| \leq 2 \sqrt{\frac{2 \sigma_X^2 \log(16n)}{n}}.$$

*(iii)* Finally, by taking $\lambda_{\mathrm{OR}}$ as in eq. (35) as well as following Lemma 3, we will have that, with probability at least $1 - 1/(2n)$,

$$\|\widehat{\alpha}_{1,\mathrm{t}} - \alpha_{1,\mathrm{t}}\|_1 + \|\widehat{\alpha}_{0,\mathrm{t}} - \alpha_{0,\mathrm{t}}\|_1 \leq 5\lambda_{\mathrm{OR}} \left( \frac{1}{2\psi^2} + \frac{2}{\psi} + \frac{s_0}{2\phi_0^2} + \frac{s_1}{2\phi_1^2} \right).$$

Now, plugging *(i), (iii), (iii)* back to the decomposition we will have that, with probability at least $1 - 1/n$, the error $|\widehat{\tau}_{\mathrm{TLDR}} - \tau|$ can be (upper) bounded by:

$$2\sqrt{\frac{2\sigma_X^2 \log(16n)}{n}} + 5M_X \left( \frac{1}{m_g} - 1 \right) \lambda_{\mathrm{OR}} \left( \frac{1}{2\psi^2} + \frac{2}{\psi} + \frac{s_0}{2\phi_0^2} + \frac{s_1}{2\phi_1^2} \right) + 2\sqrt{\frac{2\sigma^2 \log(16n)}{rnm_g^2}}$$

$$= \mathcal{O} \left( (s_0 + s_1)\sqrt{\log d} \left( \frac{1}{\sqrt{rn}} + \frac{d}{\sqrt{rn_{\mathrm{s}}}} \right) \right).$$

To get (36), we only need to notice that, for any constant $\delta > 0$, for large enough $n$ the following holds:

$$\sqrt{\log(16n)} = \sqrt{\log 16 + \log n} \leq (1 + \delta)\sqrt{\log n}.$$

We can handle the $\log(16nd)$ term in $\lambda_{\mathrm{OR}}$ (35) in a similar manner and obtain $\sqrt{\log(16nd)} \leq (1 + \delta)\sqrt{\log(nd)}$ when $n$ is large enough. Now, we obtain the non-asymptotic result in eq. (36) and complete the proof.

$\square$

# F. Additional Details of Synthetic-Data Experiments

Here, for GLM-based parametric approach, we consider IPW estimator for demonstration purpose, since the focus of our numerical simulation is on NN-based non-parametric approaches (see Section 6 and Appendix G).

## F.1. Motivating toy example

In the toy example presented in Appendix B.2, the r.v.s $X_1 \sim N(\mu_1, 1), X_2 \sim N(\mu_2, 1), \epsilon \sim N(0, 1/4)$. We choose $\mu_1 = 0, \beta_1 = 0.1$ for both domains; in target domain, we take $\mu_2 = 2, \beta_2 = -0.1$ and the causal effects are chosen as $\tau = -2/30 \approx -0.067$ and $\alpha = 0.1$; in source domain, we take $\mu_{2,\mathrm{s}} = 1, \beta_{2,\mathrm{s}} = -0.2$.

We randomly generate 2000 samples from target domain and the IPW estimate is $-0.0531$, which is pretty close to the ground truth ACE and validates the effectiveness of IPW estimator. However, in our TCL set-up, we consider limited target domain samples in that we can only observe the first 100 target domain samples, which yields a very biased IPW estimate: 0.0002. Additionally, we observe 1000 randomly generated samples from the source domain, but we "do not know" whether or not the ACEs and the treatment assignment mechanisms are the same across both domains; apparently, in our toy example, treatment assignment mechanisms are different.

In this work, we aim to leverage the abundant source domain data to improve the propensity score estimation via TL techniques. Since we do not assume same ACEs in both domains, we evaluate the IPW estimator only using the target domain data. However, fitting the PS model using the naively merged datasets (i.e., without the TL techniques) would fail since the treatment assignment mechanisms across different domains are different: in our numerical example, this naive approach yields a IPW estimate $\widehat{\tau}_{\mathrm{naive}} = 0.0441$, which is even more biased than only using target domain data. Our proposed $\ell_1$-TCL does help yield a more accurate estimated ACE: $\widehat{\tau}_{\mathrm{TLIPW}} = -0.0013$. This toy example tells us that additionally incorporate the source domain data "in a smart way" by using TL technique does improve the IPW estimator's accuracy — we can now at least infer that there is a inhibiting causal effect from treatment $Z$ to outcome $Y$.

## F.2. Synthetic-data experiments

Next, we consider GLM parametric approach. The goal is to simply demonstrate the effectiveness of the proposed $\ell_1$-TCL framework compared with two baselines: solely using target domain data for causal learning, which we call TO-CL, and naively merging both domains' datasets for causal learning, i.e., Merge-CL.

### F.2.1. EXPERIMENTAL CONFIGURATIONS AND TRAINING DETAILS

We generate synthetic data where the treatment assignment follows the GLM PS model (7) with randomly generated $d$-dimensional target domain nuisance parameters as well as $s$-sparse difference (from Gaussian distribution). We consider $d \in \{10, 20, 50, 75, 100\}$, $s \in \{1, 3, 5, 7, 10\}$, source domain sample size $n_s \in \{2000, 3000, 5000\}$ and target domain sample size $n \in \{100, 200, 500\}$ settings. For demonstration purpose, the link function is chosen to be sigmoid function and considered known a prior. This reduces the PS model fitting in the rough estimation step to naive logistic regression; in the $\ell_1$ regularized bias correction step, we use gradient descent to optimize the objective function (9) with respect to the (sparse) difference, which has in total 8000 iterations and learning rate 0.02 (which decays by half every 2000 iterations). Hyperparameter $\lambda_{PS}$ is chosen to maximize the treatment prediction area under ROC curve (AUC) on a validation (target domain) dataset with size 50.

### F.2.2. RESULTS

Figure 3 reports the difference between average (over independent 100 trials) ACE estimation errors of our proposed and the baseline frameworks: positive values indicate improved accuracy whereas negative values are all truncated to zeros for better visualization. From the comparison with TO-CL (left panel), we can observe that $\ell_1$-TCL yields more accurate ACEs for most considered experimental settings. Most importantly, we can observe that the benefit from our TL approach is the most significant when we have limited amount of target domain data and this benefit gradually disappears when we have more and more target domain data. From the comparison with Merge-CL (right panel), we can observe improved accuracy for almost all settings, verifying that Merge-CL is inherently flawed due to the the different PS models between target and source domain. In our semi-synthetic data (or pseudo-real-data) experiment, we will not consider Merge-CL.

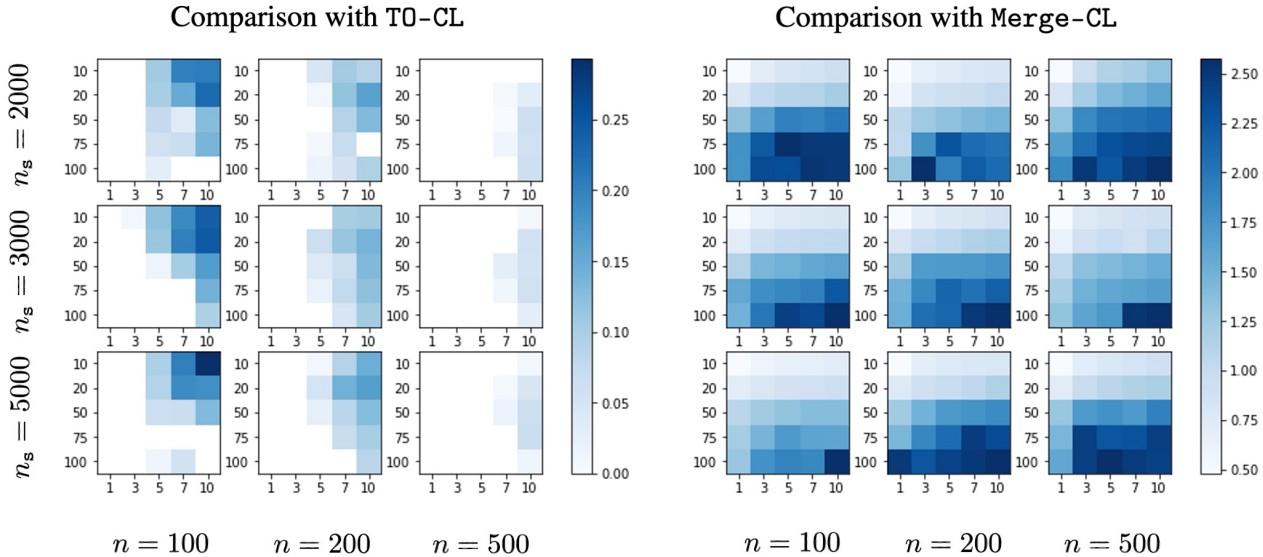

*Figure 3.* Comparison between our proposed $\ell_1$-TCL with TO-CL (left) and Merge-CL (right) baseline learning frameworks. In each sub-heatmap, x-axis represents the sparsity $s$, and the y-axis represents the dimensionality $d$. We report the difference between average ACE estimation errors of our proposed and the baseline frameworks: positive values indicate improved accuracy whereas negative values are all truncated to zeros for better visualization.

## G. Additional Details of Pseudo Real-Data Experiments

### G.1. Description of IHDP dataset

The IHDP dataset was collected from an observational study in which the goal was to observe the impact of visits from a healthcare provider on children's cognitive development. Patients were placed in the treatment group if they received special care or home visits from a provider. A semi-synthetic dataset was created from the original dataset by removing a nonrandom amount of the treatment group in order to induce treatment imbalance. The final cohort consists of 747 subjects, with 139 in the treatment group and 608 in the control group. Each subject contains 25 covariates, 6 of which are continuous

*Table 5.* List of covariates in the IHDP dataset.

| Type | Name |
|---|---|
| Child's measurements | Birth Weight, Head Circumference, Weeks Born Preterm, Birth Order, First Born, Neonatal Health Index, Sex, Twin Status. |
| Behaviors observed during pregnancy | Smoked Cigarettes, Drank Alcohol, Took Drugs |
| Mother's measurements at time of birth | Age, Marital Status, Educational Attainment, Worked During Pregnancy, Received Prenatal Care, Resident Site at Start of Intervention. |

and 19 of which are binary. These variables were collected from the child's measurements, the mother's behaviors during pregnancy, and the mother's measurements at the time of birth. The variables are detailed in Table 5.

### G.2. Experimental configurations and training details

The source-target domain partition is based on the 4-th categorical covariate, which yields $n_s = 546$ source domain sample (with labels 0) and $n = 201$ target domain samples (with labels 1). The rough estimation step in the nuisance parameter estimation stage is simply done by applying Dragonnet or TARNet fitting on the source domain data; subsequently, in the bias correction step, we randomly select 100 target domain samples as in-sample data, which are (randomly) decomposed into 70 training samples and 30 validation samples, and 101 out-of-sample testing data.

The goal is to compare three learning frameworks: `TO-CL`, `WS-TCL` and our proposed $\ell_1$-`TCL`. Additionally, we use the IPW, OR, and DR estimators for the downstream ACE estimation, which results in a total of 9 *estimation procedures*; here, we call a specific learning framework coupled with a specific ACE estimator a estimation procedure (recall that an ACE estimator is defined as a specific nuisance model coupled with a specific plug-in estimator for ACE).

Formally, the nuisance parameter estimation stage of, for example, Dragonnet-based $\ell_1$-`TCL` is done by:

$$\underline{\text{Rough estimation for Dragonnet}}: \widehat{\theta}_s = \arg\min_\theta \ \mathcal{L}_{\text{Dragon}}(\theta; \mathcal{D}_s),$$

$$\underline{\text{Bias correction for Dragonnet}}: \ \ \widehat{\theta}_t = \widehat{\theta}_s + \arg\min_\Delta \ \mathcal{L}_{\text{Dragon}}(\Delta + \widehat{\theta}_s; \mathcal{D}_t) + \lambda\|\Delta\|_1,$$

where the objective function $\mathcal{L}_{\text{Dragon}}$ is defined in eq. (17).

For `WS-TCL` as well as our proposed $\ell_1$-`TCL` frameworks, the rough estimation step uses default NN hyperparameters in the open source implementation. Notice that those hyperparameters may not be optimal for the rough estimation step using source domain data since the default hyperparameters are optimized using the full data; however, due to limited computational resources, we only consider the following hyperparameter selection based on grid-search in the bias correction step: We consider `learning rate` $\in \{$1e-6, 2e-6, 5e-6, 1e-5, 2e-5, 5e-5, 1e-4, 1e-3$\}$ and `batch size` $\in \{1, 3, 6, 16, 32, 64\}$; in particular, for our proposed $\ell_1$-`TCL`, we also consider `regularization strength` $\lambda \in \{$1e-6, 5e-6, 1e-5, 5e-5, 1e-4, 1e-3, 1e-2, 1e-1, 5e-1, 1e1$\}$. For fair comparison, we additionally optimize the hyperparameter for the baseline `TO-CL` framework by considering `learning rate` $\in \{$1e-6, 2e-6, 5e-6, 1e-5, 2e-5, 5e-5, 1e-4, 1e-3$\}$ and `batch size` $\in \{1, 3, 6, 16, 32, 64\}$. We will show that, even with sub-optimal NN hyperparameters in the rough estimation steps, the TCL frameworks outperform the baseline `TO-CL` framework, verifying the necessity of using source domain data for accurate ACE estimation.

The hyperparameter selection criteria are: NN regression loss, NN classification CE loss and MSE. Again let us take Dragonnet as an example, slightly abuse the notation and let $(\boldsymbol{x}_i, z_i, y_i)$, $i = 1, \ldots, n$ denote the validation target dataset, then the three aforementioned hyperparameter selection metrics are: $\frac{1}{n}\sum_{i=1}^n (m_{z_i}(\theta; \boldsymbol{x}_i) - y_i)^2$, $\frac{1}{n}\sum_{i=1}^n \text{CE}(e(\theta; \boldsymbol{x}_i), z_i)$ and $\frac{1}{n}\sum_{i=1}^n (e(\theta; \boldsymbol{x}_i) - z_i)^2$.

### G.3. Additional results

We report additional results using NN classification-based criteria in Table 6, where the column-wise best results are highlighted with green background color, indicating the best learning framework for the corresponding ACE estimator, and the smallest error is highlighted in bold font, indicating the overall best estimation procedure. Results in Table 6 exhibit similar patterns as shown in Table 2 that TL, especially our proposed $\ell_1$-TCL, generally helps improve ACE estimation accuracy, and IPW estimators do not perform well. Furthermore, as we can see from Table 6 In the case where WS-TCL outperforms our proposed $\ell_1$-TCL, such as the out-of-sample accuracy for TARNet-DR ACE estimator in the "NN classification CE loss" sub-table, the improvement of WS-TCL is typically marginal; in contrast, when $\ell_1$-TCL performs the best, the improvement is significant, see, e.g., the rest TARNet-OR and TARNet-DR ACE estimators in both sub-tables of Table 6 as well as Table 2.

*Table 6.* Additional mean and standard deviation table for other hyperparameter selection criteria (specified on top of each sub-table) using the same source-target domain partition as in Table 2.

Hyperparameter selected based on minimum validation NN classification CE loss

| In-sample | Dragonnet | | | TARNet | | |
| --- | --- | --- | --- | --- | --- | --- |
| | IPW | OR | DR | IPW | OR | DR |
| TO-CL | $11.411_{(30.657)}$ | $0.615_{(0.773)}$ | $0.675_{(0.649)}$ | $3.926_{(4.234)}$ | $0.768_{(0.545)}$ | $0.464_{(0.365)}$ |
| WS-TCL | $10.455_{(18.907)}$ | $0.682_{(0.75)}$ | $0.851_{(0.948)}$ | $3.418_{(3.428)}$ | $0.469_{(0.334)}$ | $0.383_{(0.262)}$ |
| $\ell_1$-TCL | $11.781_{(27.023)}$ | $0.813_{(1.123)}$ | $0.907_{(1.135)}$ | $4.012_{(6.241)}$ | $0.328_{(0.245)}$ | $\mathbf{0.326}_{(0.334)}$ |

| Out-of-sample | Dragonnet | | | TARNet | | |
| --- | --- | --- | --- | --- | --- | --- |
| | IPW | OR | DR | IPW | OR | DR |
| TO-CL | $32.228_{(45.735)}$ | $0.629_{(0.548)}$ | $1.513_{(1.627)}$ | $4.144_{(5.009)}$ | $0.806_{(0.641)}$ | $0.433_{(0.482)}$ |
| WS-TCL | $29.121_{(34.475)}$ | $0.629_{(0.816)}$ | $2.535_{(4.217)}$ | $4.371_{(5.18)}$ | $0.529_{(0.484)}$ | $\mathbf{0.349}_{(0.339)}$ |
| $\ell_1$-TCL | $33.303_{(54.411)}$ | $0.746_{(0.951)}$ | $2.639_{(3.806)}$ | $5.779_{(9.668)}$ | $0.426_{(0.33)}$ | $0.35_{(0.429)}$ |

Hyperparameter selected based on minimum validation NN classification MSE

| In-sample | Dragonnet | | | TARNet | | |
| --- | --- | --- | --- | --- | --- | --- |
| | IPW | OR | DR | IPW | OR | DR |
| TO-CL | $10.116_{(24.513)}$ | $0.767_{(1.077)}$ | $1.033_{(1.075)}$ | $4.647_{(5.418)}$ | $0.607_{(0.539)}$ | $0.474_{(0.322)}$ |
| WS-TCL | $8.342_{(13.545)}$ | $0.664_{(0.868)}$ | $0.719_{(0.789)}$ | $3.77_{(5.429)}$ | $0.589_{(0.431)}$ | $0.408_{(0.296)}$ |
| $\ell_1$-TCL | $8.249_{(13.454)}$ | $0.623_{(0.848)}$ | $0.704_{(0.759)}$ | $4.012_{(6.241)}$ | $0.328_{(0.245)}$ | $\mathbf{0.326}_{(0.334)}$ |

| Out-of-sample | Dragonnet | | | TARNet | | |
| --- | --- | --- | --- | --- | --- | --- |
| | IPW | OR | DR | IPW | OR | DR |
| TO-CL | $31.608_{(55.768)}$ | $0.743_{(0.892)}$ | $2.261_{(3.283)}$ | $4.764_{(6.154)}$ | $0.739_{(0.715)}$ | $0.383_{(0.384)}$ |
| WS-TCL | $26.398_{(47.011)}$ | $0.663_{(1.014)}$ | $1.711_{(2.249)}$ | $5.437_{(8.29)}$ | $0.73_{(0.698)}$ | $0.451_{(0.569)}$ |
| $\ell_1$-TCL | $25.786_{(46.738)}$ | $0.661_{(0.947)}$ | $1.666_{(2.185)}$ | $5.779_{(9.668)}$ | $0.426_{(0.33)}$ | $\mathbf{0.35}_{(0.429)}$ |

In addition, we consider another (randomly selected) binary covariate for the source-target domain partition, which is the 8-th categorical covariate and yields $n_s = 642$ source domain sample (with labels 0) and $n = 105$ target domain samples (with labels 1). The (random) train-validation split gives 73 training samples (that is why we choose the largest batch size grid to be 64) and 32 validation samples. We repeat the same experiments and report the results in Table 7.

In Table 7, we can observe similar patterns as mentioned before: The best in-sample performance is 0.352, which is given by TARNet-DR estimator using proposed $\ell_1$-TCL, and it is much better than the best WS-TCL in-sample performance, which is 0.375 also given by TARNet-DR estimator. In contrast, even though WS-TCL yields the best out-of-sample performance via TARNet-OR estimator (0.623), there is only a marginal increment compared to that of $\ell_1$-TCL (0.627). Additionally, the best out-of-sample performance of $\ell_1$-TCL is still given using the NN regression loss as the hyperparameter selection criterion, which is consistent with previous findings.

Lastly, since both WS-TCL and $\ell_1$-TCL are sensitive to the choice of hyperparameters, there will be cases/trials where the optimal empirical option of not covered by the pre-defined grid, leading to unfavorable results. Since our $\ell_1$-TCL has one additional hyperparameter, i.e., the regularization strength, such effect will be amplified for $\ell_1$-TCL given the limited computation resources when we perform grid search for hyperparameter selections. Therefore, we report the the median and

IQR of the aforementioned experiments using 8-th categorical covariate for source-target domain partition in Table 8, which shows that the both the in-sample and out-of-sample best (in terms of median) estimates are given by our $\ell_1$-TCL. Overall all results above support the effectiveness of our proposed $\ell_1$-TCL framework.

*Table 7.* Additional absolute error mean and standard deviation table for all aforementioned hyperparameter selection criteria (specified on top of each sub-table) using the a different (from that of Table 2) source-target domain partition.

### Hyperparameter selected based on minimum validation NN regression loss

| In-sample | Dragonnet | | | TARNet | | |
|---|---|---|---|---|---|---|
| | IPW | OR | DR | IPW | OR | DR |
| TO-CL | $5.752_{(14.189)}$ | $0.572_{(0.483)}$ | $0.592_{(0.775)}$ | $5.713_{(9.2)}$ | $0.466_{(0.406)}$ | $0.457_{(0.434)}$ |
| WS-TCL | $6.448_{(10.771)}$ | $0.572_{(0.463)}$ | $0.675_{(1.08)}$ | $1.703_{(2.018)}$ | $0.44_{(0.362)}$ | $\mathbf{0.375}_{(0.291)}$ |
| $\ell_1$-TCL | $5.376_{(9.682)}$ | $0.477_{(0.526)}$ | $0.526_{(0.884)}$ | $1.697_{(2.017)}$ | $0.448_{(0.362)}$ | $0.376_{(0.292)}$ |

| Out-of-sample | Dragonnet | | | TARNet | | |
|---|---|---|---|---|---|---|
| | IPW | OR | DR | IPW | OR | DR |
| TO-CL | $16.704_{(24.684)}$ | $0.89_{(1.05)}$ | $1.927_{(3.917)}$ | $9.916_{(12.285)}$ | $0.736_{(0.811)}$ | $0.685_{(0.851)}$ |
| WS-TCL | $21.43_{(22.895)}$ | $0.821_{(1.105)}$ | $2.252_{(2.92)}$ | $7.699_{(12.458)}$ | $\mathbf{0.623}_{(0.908)}$ | $0.723_{(1.089)}$ |
| $\ell_1$-TCL | $18.03_{(21.948)}$ | $0.731_{(1.318)}$ | $1.661_{(2.259)}$ | $7.703_{(12.46)}$ | $0.627_{(0.906)}$ | $0.724_{(1.084)}$ |

### Hyperparameter selected based on minimum validation NN classification CE loss

| In-sample | Dragonnet | | | TARNet | | |
|---|---|---|---|---|---|---|
| | IPW | OR | DR | IPW | OR | DR |
| TO-CL | $5.752_{(14.189)}$ | $0.572_{(0.483)}$ | $0.592_{(0.775)}$ | $6.302_{(12.42)}$ | $0.516_{(0.47)}$ | $0.496_{(0.703)}$ |
| WS-TCL | $6.448_{(10.771)}$ | $0.572_{(0.463)}$ | $0.675_{(1.08)}$ | $1.703_{(2.018)}$ | $0.44_{(0.362)}$ | $0.375_{(0.291)}$ |
| $\ell_1$-TCL | $6.442_{(10.761)}$ | $0.577_{(0.466)}$ | $0.677_{(1.139)}$ | $2.056_{(2.331)}$ | $0.446_{(0.449)}$ | $\mathbf{0.352}_{(0.314)}$ |

| Out-of-sample | Dragonnet | | | TARNet | | |
|---|---|---|---|---|---|---|
| | IPW | OR | DR | IPW | OR | DR |
| TO-CL | $16.704_{(24.684)}$ | $0.89_{(1.05)}$ | $1.927_{(3.917)}$ | $10.353_{(15.414)}$ | $0.878_{(1.185)}$ | $0.782_{(1.099)}$ |
| WS-TCL | $21.43_{(22.895)}$ | $0.821_{(1.105)}$ | $2.252_{(2.92)}$ | $7.699_{(12.458)}$ | $\mathbf{0.623}_{(0.908)}$ | $0.723_{(1.089)}$ |
| $\ell_1$-TCL | $21.41_{(22.875)}$ | $0.822_{(1.114)}$ | $2.272_{(2.948)}$ | $8.048_{(13.393)}$ | $0.632_{(0.809)}$ | $0.696_{(0.917)}$ |

### Hyperparameter selected based on minimum validation NN classification MSE

| In-sample | Dragonnet | | | TARNet | | |
|---|---|---|---|---|---|---|
| | IPW | OR | DR | IPW | OR | DR |
| TO-CL | $5.752_{(14.189)}$ | $0.572_{(0.483)}$ | $0.592_{(0.775)}$ | $6.302_{(12.42)}$ | $0.516_{(0.47)}$ | $0.496_{(0.703)}$ |
| WS-TCL | $6.448_{(10.771)}$ | $0.572_{(0.463)}$ | $0.675_{(1.08)}$ | $1.703_{(2.018)}$ | $0.44_{(0.362)}$ | $0.375_{(0.291)}$ |
| $\ell_1$-TCL | $6.441_{(10.761)}$ | $0.576_{(0.465)}$ | $0.677_{(1.134)}$ | $2.056_{(2.331)}$ | $0.446_{(0.449)}$ | $\mathbf{0.352}_{(0.314)}$ |

| Out-of-sample | Dragonnet | | | TARNet | | |
|---|---|---|---|---|---|---|
| | IPW | OR | DR | IPW | OR | DR |
| TO-CL | $16.704_{(24.684)}$ | $0.89_{(1.05)}$ | $1.927_{(3.917)}$ | $10.353_{(15.414)}$ | $0.878_{(1.185)}$ | $0.782_{(1.099)}$ |
| WS-TCL | $21.43_{(22.895)}$ | $0.821_{(1.105)}$ | $2.252_{(2.92)}$ | $7.699_{(12.458)}$ | $\mathbf{0.623}_{(0.908)}$ | $0.723_{(1.089)}$ |
| $\ell_1$-TCL | $21.408_{(22.875)}$ | $0.822_{(1.112)}$ | $2.271_{(2.946)}$ | $8.048_{(13.393)}$ | $0.632_{(0.809)}$ | $0.696_{(0.917)}$ |

*Table 8.* Median and IQR ([Q1, Q3]) of absolute errors of estimated causal effects ovee 50 IHDP datasets for all aforementioned hyperparameter selection criteria (specified on top of each sub-table). The source-target domain partition is the same with that of Table 7.

### Hyperparameter selected based on minimum validation NN regression loss

| In-sample | Dragonnet | | | TARNet | | |
|---|---|---|---|---|---|---|
| | IPW | OR | DR | IPW | OR | DR |
| TO-CL | $2.46_{[1.175,5.219]}$ | $0.481_{[0.244,0.657]}$ | $0.303_{[0.185,0.676]}$ | $3.097_{[2.196,5.591]}$ | $0.383_{[0.149,0.611]}$ | $0.345_{[0.163,0.617]}$ |
| WS-TCL | $3.212_{[1.411,6.828]}$ | $0.46_{[0.291,0.717]}$ | $0.433_{[0.192,0.731]}$ | $1.086_{[0.512,1.851]}$ | $0.338_{[0.165,0.662]}$ | $0.289_{[0.141,0.547]}$ |
| $\ell_1$-TCL | $2.048_{[1.251,5.817]}$ | $0.339_{[0.155,0.589]}$ | $\mathbf{0.288}_{[0.112,0.593]}$ | $1.086_{[0.511,1.784]}$ | $0.353_{[0.159,0.659]}$ | $0.289_{[0.142,0.542]}$ |

| Out-of-sample | Dragonnet | | | TARNet | | |
|---|---|---|---|---|---|---|
| | IPW | OR | DR | IPW | OR | DR |
| TO-CL | $9.297_{[3.413,14.861]}$ | $0.587_{[0.296,0.979]}$ | $0.835_{[0.277,1.876]}$ | $5.068_{[2.957,12.122]}$ | $0.451_{[0.22,0.87]}$ | $0.452_{[0.199,0.825]}$ |
| WS-TCL | $15.119_{[4.856,27.383]}$ | $0.513_{[0.286,0.901]}$ | $0.979_{[0.547,2.246]}$ | $3.636_{[0.973,10.543]}$ | $0.425_{[0.121,0.764]}$ | $0.399_{[0.157,0.897]}$ |
| $\ell_1$-TCL | $12.953_{[5.219,19.517]}$ | $\mathbf{0.368}_{[0.183,0.737]}$ | $0.862_{[0.457,1.982]}$ | $3.636_{[0.97,10.543]}$ | $0.426_{[0.127,0.762]}$ | $0.395_{[0.163,0.958]}$ |

### Hyperparameter selected based on minimum validation NN classification CE loss

| In-sample | Dragonnet | | | TARNet | | |
|---|---|---|---|---|---|---|
| | IPW | OR | DR | IPW | OR | DR |
| TO-CL | $2.46_{[1.175,5.219]}$ | $0.481_{[0.244,0.657]}$ | $0.303_{[0.185,0.676]}$ | $2.699_{[2.089,4.958]}$ | $0.359_{[0.245,0.586]}$ | $0.315_{[0.162,0.58]}$ |
| WS-TCL | $3.212_{[1.411,6.828]}$ | $0.46_{[0.291,0.717]}$ | $0.433_{[0.192,0.731]}$ | $1.086_{[0.512,1.851]}$ | $0.338_{[0.165,0.662]}$ | $0.289_{[0.141,0.547]}$ |
| $\ell_1$-TCL | $3.215_{[1.406,6.861]}$ | $0.461_{[0.298,0.719]}$ | $0.43_{[0.195,0.702]}$ | $1.187_{[0.705,2.253]}$ | $0.309_{[0.196,0.55]}$ | $\mathbf{0.27}_{[0.096,0.477]}$ |

| Out-of-sample | Dragonnet | | | TARNet | | |
|---|---|---|---|---|---|---|
| | IPW | OR | DR | IPW | OR | DR |
| TO-CL | $9.297_{[3.413,14.861]}$ | $0.587_{[0.296,0.979]}$ | $0.835_{[0.277,1.876]}$ | $5.103_{[2.116,11.363]}$ | $0.541_{[0.362,0.813]}$ | $0.51_{[0.258,0.719]}$ |
| WS-TCL | $15.119_{[4.856,27.383]}$ | $0.513_{[0.286,0.901]}$ | $0.979_{[0.547,2.246]}$ | $3.636_{[0.973,10.543]}$ | $0.425_{[0.121,0.764]}$ | $0.399_{[0.157,0.897]}$ |
| $\ell_1$-TCL | $15.117_{[4.856,27.358]}$ | $0.514_{[0.266,0.891]}$ | $0.976_{[0.593,2.275]}$ | $3.781_{[0.832,11.022]}$ | $\mathbf{0.361}_{[0.193,0.76]}$ | $0.429_{[0.201,0.748]}$ |

### Hyperparameter selected based on minimum validation NN classification MSE

| In-sample | Dragonnet | | | TARNet | | |
|---|---|---|---|---|---|---|
| | IPW | OR | DR | IPW | OR | DR |
| TO-CL | $2.46_{[1.175,5.219]}$ | $0.481_{[0.244,0.657]}$ | $0.303_{[0.185,0.676]}$ | $2.699_{[2.089,4.958]}$ | $0.359_{[0.245,0.586]}$ | $0.315_{[0.162,0.58]}$ |
| WS-TCL | $3.212_{[1.411,6.828]}$ | $0.46_{[0.291,0.717]}$ | $0.433_{[0.192,0.731]}$ | $1.086_{[0.512,1.851]}$ | $0.338_{[0.165,0.662]}$ | $0.289_{[0.141,0.547]}$ |
| $\ell_1$-TCL | $3.215_{[1.405,6.859]}$ | $0.461_{[0.299,0.719]}$ | $0.43_{[0.194,0.702]}$ | $1.187_{[0.705,2.253]}$ | $0.309_{[0.196,0.55]}$ | $\mathbf{0.27}_{[0.096,0.477]}$ |

| Out-of-sample | Dragonnet | | | TARNet | | |
|---|---|---|---|---|---|---|
| | IPW | OR | DR | IPW | OR | DR |
| TO-CL | $9.297_{[3.413,14.861]}$ | $0.587_{[0.296,0.979]}$ | $0.835_{[0.277,1.876]}$ | $5.103_{[2.116,11.363]}$ | $0.541_{[0.362,0.813]}$ | $0.51_{[0.258,0.719]}$ |
| WS-TCL | $15.119_{[4.856,27.383]}$ | $0.513_{[0.286,0.901]}$ | $0.979_{[0.547,2.246]}$ | $3.636_{[0.973,10.543]}$ | $0.425_{[0.121,0.764]}$ | $0.399_{[0.157,0.897]}$ |
| $\ell_1$-TCL | $15.116_{[4.852,27.351]}$ | $0.514_{[0.265,0.892]}$ | $0.974_{[0.593,2.275]}$ | $3.781_{[0.832,11.022]}$ | $\mathbf{0.361}_{[0.193,0.76]}$ | $0.429_{[0.201,0.748]}$ |

# H. Additional Details of Real-Data Example

## H.1. Description and pre-processing of real data

Vasopressor therapy for septic patients has been shown to decrease the risk of 28-day mortality (Avni et al., 2015). Therefore, it is worth observing the underlying causal structure of this treatment. Indeed, Wei et al. (2022) showed that vasopressor therapy may have a inhibiting effect on sepsis, which suggests its inhibiting effect on mortality as well.

In our study, all patient data for each encounter is binned into hourly windows that begin with hospital admission and end with discharge. If more than one measurement occurs in an hour, then the average of the values is recorded. To ensure that causal effect estimation is performed on data series of similar lengths (which could help control potentially unobserved confounding), patients are included in the cohort if they meet the Sepsis-3 criteria during the hospital stay, and we examine exactly 12 hours of data before and after sepsis onset, resulting in a 25-hour subset of the full patient encounter (i.e., the hour of sepsis onset as well as 12 hours before and after this time). The resulting summary statistics of the patient demographics are reported in Table 9.

*Table 9.* Summary statistics for patient demographics in selected cohort; Q1 and Q3 stand for the 25% and 75% quantiles, which gives the interquartile range (IQR).

|  | Source | | Target | |
|---|---|---|---|---|
|  | Treatment | Control | Treatment | Control |
| Number | 207 | 1249 | 58 | 700 |
| Age, median and [Q1,Q3] | $63.0$ $_{[51.0,70.5]}$ | $64.0$ $_{[53.0,74.0]}$ | $55.5$ $_{[37.25,67.5]}$ | $58.0$ $_{[41.0,68.0]}$ |
| Male, number and percentage | $131$ $_{(63.3\%)}$ | $652$ $_{(52.2\%)}$ | $38$ $_{(65.5\%)}$ | $449$ $_{(64.1\%)}$ |
| 28D-Mortality, number and percentage | $43$ $_{(20.8\%)}$ | $117$ $_{(9.4\%)}$ | $20$ $_{(34.5\%)}$ | $71$ $_{(10.1\%)}$ |
| Total Hospital Days, median and [Q1,Q3] | $22.0$ $_{[12.5,33.5]}$ | $13.0$ $_{[8.0,21.0]}$ | $25.5$ $_{[13.0,45.5]}$ | $18.0$ $_{[10.0,31.0]}$ |

The 28-day mortality, i.e., the binary outcome variables, is defined as the patient's death within 28 days or less after the time of admission. Covariates from the EMR data include:

- Vital Signs — in the ICU environment, these are usually recorded at hourly intervals. However, patients on the floor may only have measurements for every 8 hours.

- Laboratory Results — the Lab tests are most commonly ordered on a daily basis. However, the collection frequency may change based on the severity of a patient's illness and clinician's request.

Our study considers in total 4 vital signs and 30 Lab results, as presented in Table 10. Since the covariate names explain themselves, we omit further descriptions of those covariates.

*Table 10.* List of covariates, i.e., vital signs and Lab results, included in the real-data example.

| Type | Name |
|---|---|
| Vital signs | Temperature (°C), Pulse (Heart Rate), Oxygen Saturation by Pulse Oximetry (SpO2), Best Mean Arterial Pressure (MAP). |
| Lab result | Excess Bicarbonate (Base Excess), Blood Urea Nitrogen (BUN), Calcium, Chloride, Creatinine, Glucose, Magnesium, Phosphorus, Potassium, Hemoglobin, Platelets, White Blood Cell Count (WBC), Alanine Aminotransferase (ALT), Albumin, Ammonia (NH3), Aspartate Transaminase (AST), Direct Bilirubin, Total Bilirubin, Fibrinogen, International Normalized Ratio (INR), Lactic Acid (Lactate), Partial Thromboplastin Time (PTT), Prealbumin, B-type Natriuretic Peptide (BNP), Troponin I, Fraction of Inspired Oxygen (FiO2), Partial Pressure of Carbon Dioxide (PaCO2), pH, Arterial Oxygen Saturation (SaO2), Glasgow Coma Scale (GCS) Total Score. |

It is common for vital signs and laboratory results to be missing due to the recording irregularity issues listed earlier. To handle this problem, we imputed any missing values. Values were first imputed using the fill-forward method. In the fill-forward method, any missing hourly values are replaced with the most recent value from the preceding hours. Any remaining missing values were then imputed using the population median. Lastly, for each patient, we take the first data point after the time of the sepsis onset time for our experiments.

### H.2. Experimental configurations and training details

The hyperparameter of TLIPW estimator (12) in our $\ell_1$-`TCL` framework is selected via 5-fold CV using target domain data based on maximum average treatment classification prediction AUC. As mentioned previously, we use vanilla logistic regression for rough estimation using source domain data, and in the $\ell_1$ regularized bias correction step, we use gradient descent to optimize the objective function (9) with respect to the (sparse) difference. In our implementation of the bias correction step, we perform grid search over `total number of iterations` $\in \{5000, 10000, 20000\}$, `initial learning rate` $\in \{0.05, 0.02, 0.01, 0.005, 0.001\}$, `learning rate decay ratio` $\in \{0.5, 0.8, 0.9, 0.95\}$ and $\ell_1$ `regularization strength` $\log_{10} \lambda \in \{-2.5, -2.25, -2, \dots, 0\}$. The learning rate decays every 1000 iterations.

For the bootstrap uncertainty quantification, we re-fit the model with hyperparameter re-selected for each bootstrap sample; then, $90\%$ confidence interval is constructed based on the $5\%$ and $95\%$ quantiles of the bootstrap ACE estimates. Additionally, the mean and median of the bootstrap ACE estimates are reported.

