# OpenReview forum: "Transfer Causal Learning: Causal Effect Estimation with Knowledge Transfer"
_ICML.cc/2023/Workshop/IMLH — IMLH 2023 Poster_

### Official Review · Reviewer_xXGV · 2023-06-07
**The authors propose a method using Lasso regularization to correct the rough estimator using source domain data.**

**Rating:** 9
**Confidence:** 4

**Review:**

This is a well-written paper. The notations are very clear and logic is smooth. The authors propose a framework for the transfer causal learning problem, which entails data-integrative transfer learning of the nuisance parameter and plug-in estimation for causal effect in the target domain. They also establish nonasymptotic recovery guarantees for the GLM under the sparsity assumption for their proposed framework.
1. However, the authors mention that there are many methods applying data-integrative TL techniques to causal inference in the presence of heterogeneous covariate spaces, these methods fail to handle the same covariate space setting. It is unclear to me why handling same covariate space is more difficult, compared to heterogeneous covariate spaces. It would be better if the authors could explain this difference when they demonstrate their method.
2. The paper is based on the assumption of the sparsity of the nuisance parameter difference. This looks like a big assumption to me, since it is rare that the differences are zeros. The sparsity of the coefficients themselves ($\beta_t$ and $\beta_s$) is more natural. The authors should demonstrate the practicality of this assumption or draw the connection between sparsity of the coefficients and the sparsity of their difference.
3. There are a few possible typos. (i) In equation (12), should $\alpha_{1,t}$ and $\alpha_{0,t}$ in $\tau_{TLOR}$ be with hat? (ii) In the right column of line 298, the authors say "we will use the formulation in eq.(9)...". Since this is the section for a generic framework, it should not be eq.(9) from GLM. Plus, in Appendix G2, they use $L_{Dragon}$ instead.

---

### Meta-Review · Area_Chair_zBQb · 2023-06-20

**Recommendation:** Accept (Poster)
**Confidence:** 3

**Metareview:**

This work proposes a l1-TCL method that incorporates l1 regularizer for parameter estimation in transfer causal learning. The experiments on the pseudo and real data shows the effectiveness of the method in estimating the treatment effect. The method is solid with no identified major flaws. This paper is well-written.

Some minor comments:

In the pseudo and real data experiments, how did covariates in X determined to fulfill the Ignorability Assumption?

Does TCL have the assumption that features in X should be the same? For example, due to the difference in causal relationship, a feature in one domain should be controlled but doesn't need to be controlled in another domain. If so, as stated in "treatment assignment mechanisms should be very similar across both domains", such assumption for the general proposed method should be made explicit.

Please clarify the meaning of unbiased, as the word is overloaded and means differently in causal inference and in ML fairness.

"RCT is unethical in most studies, such as medical study. "
Since RCT is the level 1 evidence in the evidence-based medical practice, please revise the statement about the shortcomings of RCT accordingly.

---

### Decision · Program_Chairs · 2023-06-20

Accept (Poster)